# Luminescent Lanthanide Infinite Coordination Polymers for Ratiometric Sensing Applications

**DOI:** 10.3390/molecules30020396

**Published:** 2025-01-18

**Authors:** Ziqin Song, Yuanqiang Hao, Yunfei Long, Peisheng Zhang, Rongjin Zeng, Shu Chen, Wansong Chen

**Affiliations:** 1Key Laboratory of Theoretical Organic Chemistry and Functional Molecule of Ministry of Education, School of Chemistry and Chemical Engineering, Hunan University of Science and Technology, Xiangtan 411201, China; songziqin1213@163.com (Z.S.); lyf@hnust.edu.cn (Y.L.); pshzhang07@gmail.com (P.Z.); zrjxh2@126.com (R.Z.); chenshu@hnust.edu.cn (S.C.); 2College of Chemistry and Chemical Engineering, Central South University, Changsha 410017, China

**Keywords:** lanthanide luminescent materials, infinite coordination polymers, ratiometric sensing

## Abstract

Ratiometric lanthanide coordination polymers (Ln-CPs) are advanced materials that combine the unique optical properties of lanthanide ions (e.g., Eu^3+^, Tb^3+^, Ce^3+^) with the structural flexibility and tunability of coordination polymers. These materials are widely used in biological and chemical sensing, environmental monitoring, and medical diagnostics due to their narrow-band emission, long fluorescence lifetimes, and excellent resistance to photobleaching. This review focuses on the composition, sensing mechanisms, and applications of ratiometric Ln-CPs. The ratiometric fluorescence mechanism relies on two distinct emission bands, which provides a self-calibrating, reliable, and precise method for detection. The relative intensity ratio between these bands varies with the concentration of the target analyte, enabling real-time monitoring and minimizing environmental interference. This ratiometric approach is particularly suitable for detecting trace analytes and for use in complex environments where factors like background noise, temperature fluctuations, and light intensity variations may affect the results. Finally, we outline future research directions for improving the design and synthesis of ratiometric Ln-CPs, such as incorporating long-lifetime reference luminescent molecules, exploring near-infrared emission systems, and developing up-conversion or two-photon luminescent materials. Progress in these areas could significantly broaden the scope of ratiometric Ln-CP applications, especially in biosensing, environmental monitoring, and other advanced fields.

## 1. Introduction

Lanthanide luminescent materials are a class of functional materials with unique optical properties, playing a crucial role in various fields such as optoelectronics, sensors, display technology, and biomedicine [1,2,3,4,5,6,7,8]. These materials are typically formed by coordinating rare earth metal ions (e.g., Eu^3+^, Tb^3+^, Nd^3+^, Er^3+^) with inorganic or organic ligands, resulting in compounds with distinct optical characteristics. The main features of lanthanide luminescent materials include narrow-band emission, high spectral purity, and long-lived fluorescence, offering significant advantages in applications such as spectroscopy, time-resolved spectroscopy, and detection. Furthermore, these materials exhibit excellent resistance to photobleaching and high sensitivity, enabling precise responses to environmental changes such as temperature, pH, and ion concentration, making them widely applicable in biosensors and chemical sensors [9,10,11,12,13,14,15,16]. Lanthanide luminescent materials also show great potential in energy conversion [17,18], catalysis [19,20], drug delivery [21,22,23], biological imaging and disease diagnosis [24,25]. Based on their chemical composition and structure, lanthanide luminescent materials can be categorized into lanthanide-doped materials, lanthanide coordination compounds, lanthanide polymers, and lanthanide composite materials. Each category demonstrates unique advantages in specific applications, particularly in sensors, optoelectronic devices, and energy conversion. With the advancement of nanotechnology, the functionality and application range of lanthanide luminescent materials are expected to continue expanding, making them an essential component of modern technological innovations and research.

Among lanthanide luminescent materials, infinite coordination polymers (ICPs), also known as amorphous coordination polymers (ACPs), are a class of unique amorphous materials with distinct advantages [26,27,28,29,30,31]. These materials are widely applied in sensors [32,33,34,35], catalysis [36,37,38], biological imaging [39,40], drug delivery [41,42], and other fields [43,44,45]. Compared to crystalline coordination polymers and metal–organic frameworks (MOFs), ICPs lack long-range order, exhibiting greater flexibility and adaptability. This enables them to respond more effectively to environmental changes such as temperature, humidity, and solvent variations, making them particularly suitable for applications requiring flexible responses, such as biological and chemical sensors. In addition, ICPs demonstrate excellent resistance to photobleaching, maintaining a stable fluorescence intensity under prolonged light exposure, which gives them a unique advantage in long-term monitoring and high-sensitivity detection. Compared to crystalline materials, ICPs offer greater advantages in synthesis and processing. They can be synthesized under milder conditions and are easily processed into films, coatings, or composite materials, reducing the preparation complexity and cost. This makes ACPs more suitable for large-scale production and practical applications. The amorphous structure of ICPs also provides greater flexibility in functionalization, allowing different functional groups to be easily introduced according to needs. This expands their potential for multifunctional applications in sensors, catalysis, and drug delivery.

In recent years, luminescent lanthanide infinite coordination polymers, hereafter referred to as lanthanide coordination polymers (Ln-CPs), have emerged as powerful tools in analytical detection. Composed of rare earth metal ions (such as Eu^3+^, Tb^3+^, etc.) coordinated with organic ligands, Ln-CPs are known for their narrow-band emission, high spectral purity, long fluorescence lifetime, and excellent resistance to photobleaching. These properties make them highly suitable for applications in biosensors, chemical sensors, and environmental monitoring. Ratiometric fluorescence sensors have significant advantages over traditional single-emission methods [46,47,48,49,50,51]. By measuring the ratio of fluorescence intensities at different wavelengths, these sensors offer self-calibration, reducing the influence of environmental factors such as light intensity, temperature, and background noise. This enhances measurement accuracy, reliability, and sensitivity, particularly for the detection of trace target analytes. Moreover, Ln-CPs exhibit strong adaptability and can effectively integrate secondary luminescent center ligands, facilitating the construction of ratiometric sensor systems [32]. Ratiometric sensors can operate in complex biological or environmental samples, making them ideal for applications in medical diagnostics for disease biomarker detection, environmental pollutant monitoring, and food safety for product quality assurance. While many reviews have discussed the analytical applications of lanthanide luminescent materials [26,52,53,54,55,56,57,58], there is still a lack of comprehensive reviews specifically focused on ratiometric lanthanide coordination polymer sensors. Therefore, this review aims to systematically present the progress of research on various ratiometric lanthanide coordination polymer probes. The paper first introduces the composition and design principles of ratiometric lanthanide coordination polymer probes, followed by a categorized discussion of their applications in the analysis of different target analytes, including ions, small biomolecules, drug molecules, and macromolecules. Finally, we provide an outlook on the current status and future development of light-switch probes, hoping that this review will offer a clear overview of ratiometric lanthanide coordination polymer probes and provide guidance and reference for the design and application of new, efficient fluorescence probes.

## 2. Ratiometric Ln-CPs: Composition and Sensing Mechanism

Ratiometric lanthanide coordination polymers (Ln-CPs) are a class of advanced materials that combine the unique optical properties of lanthanide ions with the structural flexibility and tunability of coordination polymers. These materials offer several advantages, such as high sensitivity, selectivity, and self-calibration, making them widely used in analytical detection applications.

### 2.1. Composition of Ratiometric Ln-CPs

Ratiometric Ln-CPs are normally composed of three key components: lanthanide metal ions, organic ligands, and a second luminescent center. Each plays an essential role in the structure and functionality of the material (Figure 1).

Lanthanide metal ions serve as the central luminescent component of these materials. Examples include Tb^3+^ (Terbium), Eu^3+^ (Europium) and Ce^3+^ (Cerium), among others. Lanthanide ions are selected for their unique optical properties, which include sharp emission bands arising from f-f transitions, long fluorescence lifetimes, and high sensitivity to environmental changes. These properties make lanthanide ions ideal candidates for sensing applications, where high precision and stability are essential. Their ability to emit sharp, narrow-band fluorescence without interference from background noise adds to their suitability for complex analytical applications.

The organic ligands in Ln-CPs typically function as chelating agents that coordinate with the lanthanide ions, stabilizing them within the polymer network. Common organic ligands include carboxylates, phosphates, and nitrogen-containing compounds. These ligands not only stabilize the metal center but also contribute to the photophysical properties of the material by facilitating energy transfer. For example, nucleotides or other organic molecules can be used as ligands to enhance the emission properties of the material and promote efficient energy transfer to the lanthanide ions, thereby increasing the luminescence of the material.

The introduction of a second luminescent center is what allows the Ln-CPs to function as a ratiometric sensing system. This second center could be an organic small molecule (such as coumarin or luminol), quantum dots, or even a second lanthanide ion. The second luminescent center is carefully chosen so that it emits at a different wavelength to the primary lanthanide emission. The inclusion of this second luminescent center enables the creation of two distinct emission bands: one serves as a reference signal, while the other acts as the sensing signal. The intensity ratio between these two emission bands changes in response to the concentration of the target analyte, allowing the precise measurement of the analyte’s concentration.

In certain ratiometric Ln-CPs, the reference signal (from the second luminescent center) remains relatively constant, while the sensing signal (from the lanthanide ion) fluctuates in response to environmental conditions or the presence of the target analyte. This creates a self-calibrating mechanism, where the ratio of the intensities of the two emission bands compensates for fluctuations in factors like temperature, light intensity, and background noise. This self-calibration ensures accurate and reliable measurements, making ratiometric Ln-CPs highly valuable in complex environments, such as biological and chemical sensing applications. Furthermore, the use of a second luminescent center allows for tunable emission properties, enhancing the versatility of the material for different sensing applications. The second luminescent center could even be another lanthanide ion, enabling the creation of dual emission channels from the same polymeric framework. This dual-emission capability improves the sensitivity and complexity of the ratiometric system, offering enhanced detection capabilities for a wide range of analytes.

### 2.2. Sensing Mechanism of Ratiometric Ln-CPs Sensors

The sensing mechanism of ratiometric Ln-CPs is primarily based on the variation in the emission intensity ratio of two distinct emission peaks under specific environmental conditions or in the presence of an analyte. The key factor influencing this ratio is the interaction between the target analyte and one of the components in the Ln-CP, which affects the energy transfer process between the lanthanide ions and the organic ligands or guest molecules, or alters the photophysical properties of the second luminescent center. The mechanism can be divided into the following categories (Figure 2):

(i) Sensitization effect of the target analyte on the lanthanide ion

One of the common mechanisms implicated in ratiometric Ln-CP sensing is the sensitizing effect of the target analyte on the lanthanide ion. For example, dipicolinic acid (DPA) has a sensitizing effect on Tb^3+^ ions, and tetracycline (Tc) can sensitize Eu^3+^. In this case, the fluorescence properties of the second luminescent center remain unchanged, and as the target analyte binds with the lanthanide ion, the emission intensity ratio between the two emission peaks increases. This change in the intensity ratio allows for the detection of the target analyte through a ratiometric fluorescence response.

(ii) Reaction between the analyte and the ligand or guest molecules in the Ln-CPs

Another common sensing mechanism occurs when the target analyte reacts with the ligands or guest molecules within the Ln-CPs, affecting the energy transfer process between the lanthanide ion and the organic ligand. For instance, in Tb-GMP CPs (formed with GMP as the ligand), the target analyte alkaline phosphatase (ALP) can hydrolyze GMP, disrupting the CPs structure and affecting the fluorescence of Tb^3+^ ions. The second luminescent center’s fluorescence remains unchanged, providing a ratiometric fluorescence response. Similarly, analytes such as H_2_S or Aβ peptides can interact with guest molecules, such as Cu^2+^, effectively quenching the Cu^2+^ fluorescence and leading to the restoration of the lanthanide ion’s fluorescence. This type of ratiometric response offers high sensitivity and specificity for the detection of analytes.

(iii) Reaction between the analyte and the guest molecule (second luminescent center)

In some cases, the target analyte can react with the guest molecules (or the second luminescent center), resulting in the formation of strong fluorescent products. For example, reactive oxygen species (i.e., H_2_O_2_ and HO·) can react with the guest molecules in oxidized Ln-CPs, such as terephthalic acid, converting it to 2-hydroxyterephthalic acid, a compound with strong fluorescence. This reaction leads to a ratiometric fluorescence response, where the increase in fluorescence intensity from the product can be used to quantify the target analyte.

In addition to the mechanisms implicated in the ratiometric fluorescence response beyond the interactions described above, ratiometric Ln-CPs can also undergo changes in their optical properties in response to environmental stimuli such as temperature, pH, or ion concentration. These changes can be utilized to design ratiometric sensors with high precision, as the fluorescence response will vary with these environmental factors.

## 3. Applications of the Ratiometric Ln-CPs

Ratiometric lanthanide coordination polymers (Ln-CPs) are advanced materials that combine the unique optical properties of lanthanide ions with the structural flexibility and tunability of coordination polymers. These materials offer numerous advantages, such as high sensitivity, selectivity, and self-calibration capabilities, making them widely applicable for the detection of various target species, including ions, small biomolecules, small molecule drugs, biomacromolecules, and others. These features make Ln-CPs highly valuable for chemical sensing, medical diagnostics, environmental monitoring, and biochemical research. The following sections will discuss the applications of these probes, categorized by target type, and explore their composition, operating principles, and mechanisms.

### 3.1. Ions

#### 3.1.1. Fe^2+^/Fe^3+^

Fe^2+^ is an essential component of hemoglobin, playing a critical role in oxygen transport by facilitating oxygen binding and release [59,60]. Additionally, Fe^2+^ participates in electron transfer within cells, particularly in the electron transport chain, which is crucial for cellular respiration. Fe^3+^, often found in the form of iron oxide, has a lower capacity for oxygen transport but is indispensable for the catalytic activity of various enzymes, especially those involved in iron storage and redox reactions. Imbalances in iron levels, such as iron deficiency or iron toxicity, can lead to a range of health problems, including anemia and a weakened immune system. Moreover, Fe^2+^ and Fe^3+^ are significant indicators in water quality monitoring, as they participate in the cycling of soil and water. Fe^2+^ is more stable in aquatic environments, while Fe^3+^ tends to participate in oxidative conditions. The variation between Fe^2+^ and Fe^3+^ is often used to indicate the presence of pollutants, making their detection highly important.

Li and Ye et al. introduced a ratiometric fluorescence sensor based on carbon dots-doped lanthanide coordination polymers (CDs-doped Ln-CPs) for the detection of Fe^2+^/Fe^3+^ [61]. The sensor probe is composed of the following components: carbon dots (CDs), adenosine monophosphate (AMP), the lanthanide element Tb^3+^, and the auxiliary ligand phenanthroline (Phen). In this system, carbon dots are encapsulated within the polymer structure to provide a stable reference fluorescence signal. Phen enhances the luminescence of Tb^3+^ through energy transfer. However, the characteristic emission of Tb^3+^ is quenched in the presence of Fe^3+^ due to the high affinity between Fe^2+^ and Phen, which significantly suppresses the fluorescence of Tb^3+^. This probe offers high sensitivity and self-calibration, as the carbon dots serve as an internal control to eliminate the errors caused by instrumentation or environmental changes, thereby providing accurate quantitative results. The probe also demonstrates high selectivity and is highly suitable for Fe^2+^/Fe^3+^ detection in environmental monitoring, showing promising practical applications. This research group further utilized the reducing property of ascorbic acid (AA), which can reduce Fe^3+^ to Fe^2+^, and integrated this with the CDs-doped Ln-CPs system to achieve the efficient detection of ascorbic acid [62].

#### 3.1.2. Cu^2+^

Cu^2+^ plays a crucial role both in biological systems and the environment. It is the active center of many enzymes, particularly in redox reactions, where it is involved in cellular respiration and energy generation [63,64]. Copper also plays an essential role in nerve transmission, iron metabolism, and immune responses. However, an imbalance in copper ion concentration can lead to health issues. Excessive copper ions can cause toxicity, leading to liver and kidney damage, while insufficient copper can result in anemia and reduced immune function. In addition, the detection of Cu^2+^ in the environment is equally important, as high levels of copper ions are often indicative of water pollution, which may originate from industrial wastewater or agricultural runoff. Therefore, the precise monitoring of copper ions in water is critical for water quality protection and pollution management.

Qiu et al. proposed a ratiometric fluorescence detection method for Cu^2+^ using a dual-ligand lanthanide fluorescence probe (Luminol-Tb-GMP coordination polymer nanoprobe, CPNPs) (Figure 3) [65]. The Luminol-Tb-GMP CPNPs are composed of Tb^3+^ as the central metal ion, with GMP ligands and luminol acting as the bridging ligands. In the Luminol-Tb-GMP CPNPs, luminol provides a stable fluorescence signal (at 430 nm), while the Tb^3+^ fluorescence (at 547 nm) is used to respond to the presence of Cu^2+^. In the presence of Cu^2+^, it binds with the luminol/GMP units, leading to the quenching of the GMP-Tb fluorescence (response signal), while the luminol fluorescence remains constant as the reference signal. This change enables the ratiometric fluorescence detection of Cu^2+^ by monitoring the fluorescence ratio (F_430_/F_547_). The sensitivity is significantly enhanced because each CPNP contains a large number of luminol/GMP units, and when Cu^2+^ binds with these units, the nearby Tb^3+^ fluorescence is also quenched, amplifying the change in the response signal. The detection limit for Cu^2+^ using this ratiometric fluorescence sensor is as low as 4.2 nM, which is approximately three orders of magnitude lower than that of the single ligand GMP-Tb sensor. This sensor is particularly suitable for environmental monitoring, especially for the rapid detection of copper ions in water.

#### 3.1.3. Ag^+^

Ag^+^ exhibits significant antimicrobial properties and is widely used as a disinfectant and antimicrobial agent, particularly in medicine and water treatment [66,67]. However, the accumulation of silver ions in the environment can lead to pollution problems. Elevated concentrations of silver ions can be toxic to aquatic organisms and other ecosystems, severely degrading water quality. Therefore, monitoring Ag^+^ concentrations in water and the environment is essential for the timely identification of pollution sources, as well as for ensuring environmental protection and maintaining ecological balance.

Xu et al. proposed a ratiometric fluorescence sensor for Ag^+^ detection based on a Tb^3+^-nucleotide-based coordination polymer [68]. The nanoprobe consists of three components: luminol, Tb^3+^, and GMP; together, these form the core structure of the sensor. The molecular composition is optimized to achieve a ratiometric fluorescence response. The sensor displays dual-emission characteristics: it combines the fluorescence emissions of luminol and Tb^3+^, with Ag^+^ sensitizing the fluorescence signal of Tb^3+^. By monitoring the fluorescence at different wavelengths, the concentration of Ag^+^ can be accurately quantified. When Ag^+^ is present, it interacts with specific sites in the nucleotide molecule, altering the fluorescence properties and causing a change in the fluorescence intensity. By simultaneously monitoring the fluorescence signals at two distinct wavelengths (dual-emission), the fluorescence intensity ratio (F_545_/F_425_) can be calculated, enabling the rapid and precise ratiometric fluorescence detection of Ag^+^. This method offers high sensitivity and selectivity, making it ideal for detecting silver ions in low concentration ranges. The sensor can detect Ag^+^ concentrations as low as the nM level with high accuracy, making it highly suitable for environmental monitoring, particularly for the rapid detection of silver ions in water.

#### 3.1.4. Hg^2+^

Mercury ions (Hg^2+^) are highly toxic and pose significant threats to both the environment and human health [69,70,71]. Mercury and its compounds accumulate in aquatic systems, causing severe risks to aquatic organisms and entering the food chain. High concentrations of Hg^2+^ can lead to neurotoxicity, kidney damage, and other serious health issues in both humans and wildlife. Therefore, detecting Hg^2+^ in environmental samples (such as water and soil) and biological samples is crucial for preventing mercury poisoning and monitoring pollution levels.

Li and Ye et al. proposed a ratiometric fluorescence sensor based on dye-doped lanthanide coordination polymer particles (CDs-doped Ln-CPs) for the rapid detection of mercury ions [72]. The sensor consists of three main components: AMP, Ce^3+^, and Tb^3+^ ions; these form the Ce/Tb-AMP coordination polymer through self-assembly, along with the fluorescent dye coumarin encapsulated in the assembly process. Under 310 nm excitation, the coumarin@Ce/Tb-AMP complex emits green fluorescence from Tb^3+^, accompanied by weak fluorescence at 445 nm from the coumarin. In the presence of Hg^2+^, the complex’s structure is disrupted due to the specific coordination interaction between Hg^2+^ and AMP, leading to the release of coumarin and the enhancement of its fluorescence, while Tb^3+^ fluorescence is quenched. By monitoring the fluorescence ratio of coumarin (445 nm) to Tb^3+^ (545 nm), the precise detection of Hg^2+^ is achieved. The sensor shows high sensitivity, with a detection limit of 0.03 nM and a wide linear range from 0.08 nM to 1000 nM. It was successfully applied to water samples and human blood serum, demonstrating its potential for environmental monitoring and health diagnostics. The dual-emission mechanism (coumarin and Tb^3+^ fluorescence) overcomes the limitations of traditional fluorescence sensors, providing a highly selective, self-calibrating, and practical method for Hg^2+^ detection. The research group achieved the ratiometric fluorescence detection of Hg^2+^ using a dual-ligand fluorescent probe GMP–Tb–luminol coordination polymer, which enables the highly sensitive detection of Hg^2+^ by monitoring the fluorescence intensity ratio between luminol and Tb^3+^, with excellent selectivity and a low detection limit [73].

Liang and Qiu also proposed another ratiometric fluorescence sensor based on dye-doped lanthanide coordination polymer particles, which combines luminol and isophthalic acid (IPA) as dual ligands coordinated with Eu^3+^ ions [74]. Through a self-assembly process, luminol and IPA form a supramolecular network coordinated with Eu^3+^. In this network, IPA and luminol undergo photoinduced electron transfer (PET), which quenches the fluorescence of Eu^3+^ while enhancing the blue fluorescence of luminol. When Hg^2+^ is present, it strongly interacts with the nitrogen atom in the luminol molecule, quenching its fluorescence, while disrupting the PET effect and increasing the red fluorescence from Eu^3+^. By monitoring the fluorescence ratio between Eu^3+^ (617 nm) and luminol (430 nm), the sensor achieves the highly sensitive and selective detection of Hg^2+^. The sensor has a linear detection range of 0.05 μM to 20 μM and a detection limit of 13.2 nM, with excellent selectivity that avoids interference from other metal ions. The sensor has been successfully used to detect Hg^2+^ in drinking water and human blood serum, demonstrating its significant potential in environmental monitoring and health diagnostics. Huang and Li et al. developed a dual-ligand fluorescence probe based on terbium–organic gels (Tb-L_0.2_P_0.8_ MOGs) for efficient Hg^2+^ detection [75]. The sensor was synthesized using a simple room-temperature gelation method, resulting in a 2D gel structure that displays dual-emission fluorescence from luminol and Tb^3+^ at 424 nm and 544 nm, respectively. In the presence of Hg^2+^, the luminol fluorescence at 424 nm is quenched due to Hg^2+^ coordination, which disrupts energy transfer and induces a photoinduced electron transfer (PET) effect. Meanwhile, the fluorescence of Tb^3+^ at 544 nm increases, providing a ratiometric response to Hg^2+^. The sensor exhibits a linear detection range of 0.1–30 μM for Hg^2+^ with a detection limit of 3.6 nM. Additionally, the sensor can efficiently adsorb Hg^2+^ from water, offering potential for Hg^2+^ removal in environmental water treatment. The use of mixed ligands significantly enhances the fluorescence response, making the sensor suitable for real-world applications in environmental monitoring, such as detecting Hg^2+^ in tap water and porphyra. 

#### 3.1.5. PO_4_^3−^

Phosphate ions (PO_4_^3−^) are essential in biological and environmental systems, playing a key role in energy transfer, DNA/RNA synthesis, and enzyme regulation [76,77,78,79,80]. They are critical for both plant and animal life, particularly in the formation of ATP and in cellular signaling. However, excess phosphate in the environment, often due to agricultural runoff and fertilizer use, can lead to eutrophication in water bodies, resulting in algae overgrowth, oxygen depletion, and water quality degradation. Therefore, the detection of phosphate ions is crucial for environmental monitoring, especially in wastewater treatment and agricultural management. Traditional methods, such as colorimetric assays and electrochemical sensors, are commonly used for phosphate detection, but recent advances have focused on fluorescence-based sensors due to their higher sensitivity and low detection limits.

Liu et al. developed a novel fluorescence probe for the precise detection of phosphate ions (Pi) using a dual-emission lanthanide-based system [81]. This system combines Eu^3+^, luminol and 2,6-pyridinedicarboxylic acid (DPA), where DPA facilitates energy transfer to Eu^3+^, enhancing its fluorescence and reducing the quenching effects from water molecules. The luminol component, with its inherent blue fluorescence, acts both as a stable reference and further boosts the overall fluorescence. The probe allows for ratiometric detection by monitoring the fluorescence intensity ratio between Eu^3+^ (615 nm) and luminol (423 nm), which is directly proportional to the concentration of Pi. This method offers a detection range of 0.1 to 25 µM and a detection limit of 0.027 µM, ensuring high sensitivity for both environmental and biological applications. Additionally, the probe displays a clear fluorescence color change from blue to red under UV light, facilitating visual detection. The system is also coupled with a smartphone-based platform for semi-quantitative Pi measurement, making it portable and user-friendly. The probe has been successfully applied for detecting Pi in human urine and tap water, demonstrating its practical use in environmental monitoring and health diagnostics, thereby offering a fast, simple, and reliable method for phosphate ion analysis in diverse applications. Based on the sensitizing effect of phosphate ions (Pi) on the fluorescence of Eu^3+^, Lai and Sun et al. developed a rapid and sensitive method for detecting phosphate ions (Pi) using stimulus-responsive lanthanide infinite coordination polymer nanoparticles (COU@Eu-ICPs) (Figure 4) [82]. The sensor employed a dual-emission fluorescence system, with Eu^3+^ emitting at 615 nm and coumarin providing the reference fluorescence at 423 nm. Upon the addition of Pi, the fluorescence of Eu^3+^ significantly increased, enabling ratiometric detection where the ratio of F_615_/F_423_ directly correlated with the Pi concentration. Furthermore, as the Pi concentration increased, the fluorescence color of the probe changed from blue to red, providing a visible color change that could be observed with the naked eye. A portable test strip based on this probe was successfully developed and used for the on-site detection of Pi in human urine and serum samples.

Liu et al. proposed a novel method for the sensitive and selective detection of phosphate ions using lysine-sensitized terbium coordination polymer nanoparticles (AMP-Tb/Lys) [83]. The AMP-Tb/Lys nanoparticles were composed of adenosine monophosphate (AMP) and terbium ions (Tb^3+^), with lysine (Lys) acting as a sensitizer through the antenna effect. This sensitization enhanced the luminescence of Tb^3+^ at 488 nm and 544 nm, while the Lys luminescence at 375 nm was quenched due to energy transfer from Lys to Tb^3+^. The interaction of Pi with the AMP-Tb/Lys complex resulted in the quenching of Tb^3+^ fluorescence at 488 nm and 544 nm, while the Lys fluorescence at 375 nm was enhanced, enabling ratiometric luminescence detection. The ratio of fluorescence intensities at 544 nm and 375 nm (I_544_/I_375_) was directly related to the Pi concentration. This method offered a detection range of 0.1 to 6.0 µM and a detection limit of 0.08 µM, making it suitable for the highly sensitive detection of Pi in both environmental and biological samples. The dual-emission reverse-change ratio luminescence method reduced systematic errors, minimized background luminescence interference, and ensured high selectivity and accuracy. The AMP-Tb/Lys nanoparticles were successfully applied to detect Pi in real water samples, such as river water and lake water, with acceptable recoveries. The method was also found to be highly effective for detecting Pi in human urine and serum samples.

#### 3.1.6. ^•^OH

Hydroxy radicals (^•^OH) are a member of the reactive oxygen species (ROS) family and are highly reactive molecules produced in biological systems and the environment through various metabolic and chemical processes [84,85,86]. Due to their strong oxidizing properties, hydroxy radicals are among the most reactive species, capable of causing severe damage to cellular structures, such as lipids, proteins, DNA, and cell membranes, leading to oxidative stress. Oxidative stress is linked to the development of various diseases, including cancer, neurodegenerative disorders, and cardiovascular diseases. Because ^•^OH has a short lifespan and strong reactivity, sensitive and specific detection methods are required. Traditional techniques for detecting ^•^OH include electron spin resonance spectroscopy, chromatography, and fluorescence methods. Among these, fluorescence detection is highly favored due to its simplicity, high sensitivity, and ability to provide real-time analysis.

Tan et al. developed a novel ratiometric fluorescence probe based on functionalized europium(III) coordination polymer nanoparticles (Eu/DPA-TA) for the sensitive and selective detection of hydroxy radicals [87]. The probe is composed of dipicolinic acid (DPA), which sensitizes Eu^3+^ fluorescence, and terephthalic acid (TA), a functional ligand that reacts specifically with ^•^OH. In the absence of ^•^OH, Eu/DPA emits red fluorescence at 615 nm, while TA is non-fluorescent. Upon exposure to ^•^OH, TA is converted into 2-hydroxyterephthalic acid (TAOH), which emits blue fluorescence at 445 nm. This conversion results in a ratiometric fluorescence change, and the I_544_/I_375_ ratio is directly related to the ^•^OH concentration, enabling accurate detection. The method has a detection range of 0.1 to 6.0 µM and a detection limit of 0.08 µM, providing high sensitivity and reliability. The system can also visually detect ^•^OH at concentrations as low as 10 µM, offering a practical tool for on-site monitoring. The probe was successfully applied to detect ^•^OH in real water samples and biological fluids, including human urine and serum, demonstrating its potential for environmental monitoring and medical diagnostics.

### 3.2. Small Biomolecules

#### 3.2.1. Hydrogen Peroxide

Hydrogen peroxide (H_2_O_2_) is a common and strong oxidizing agent, and is widely used in various fields such as medicine, industry, and biology [88,89,90]. It is a colorless liquid at room temperature and can rapidly decompose into water (H_2_O) and oxygen (O_2_), a reaction that is accelerated in the presence of catalysts. In biological systems, H_2_O_2_ also plays a crucial role, particularly in cell signaling and immune responses. Through catalytic breakdown by peroxidases, H_2_O_2_ is prevented from accumulating in the body, thus protecting cells from oxidative damage. As a result, H_2_O_2_ has become an important target for detection in the field of biosensors.

Liu et al. developed a novel ratiometric luminescent probe based on lanthanide coordination polymers for the sensitive detection of H_2_O_2_ in food samples [91]. The probe was constructed using adenosine monophosphate (AMP) as a bridging ligand, terbium (Tb^3+^) as the metal ion, and 3-carboxyphenylboronic acid (3-CPBA) as the sensitizer and functional ligand. The system, labeled AMP-Tb/3-CPBA, utilized the antenna effect of 3-CPBA to enhance the luminescence of Tb^3+^ at 488 nm and 544 nm. Upon adding H_2_O_2_, the non-luminescent 3-CPBA was converted into 3-hydroxybenzoic acid (HBA), which emitted strong fluorescence at 401 nm. As a result, the fluorescence intensity of AMP-Tb/3-CPBA at 544 nm decreased, while the intensity at 401 nm increased. The ratio between the luminescence intensities at 401 nm and 544 nm (I_401_/I_544_) was strongly correlated with the H_2_O_2_ concentration, allowing for precise and sensitive detection. The sensor exhibited a detection range from 0.1 to 60.0 μM and a detection limit of 0.23 μM. The ratiometric luminescence method enabled the detection of H_2_O_2_ in food samples while minimizing interference from background luminescence, making the probe highly selective and accurate. This study provided a simple, cost-effective, and reliable way to monitor H_2_O_2_ residues in food, addressing concerns related to food safety, especially the potential adverse health effects of H_2_O_2_ contamination.

Liu et al. developed a novel ratiometric fluorescent sensor based on copper-doped lanthanide coordination polymers (Ln-CPs) for the sensitive detection of H_2_O_2_ and glutathione (GSH) [92]. The sensor was fabricated by self-assembling guanine diphosphate (GDP), terephthalic acid (TA), Tb^3+^, and Cu^2+^, resulting in Tb/Cu-GDP/TA coordination polymers (CPs). The introduction of Cu^2+^ significantly enhanced the peroxidase-like activity of the CPs compared to free Cu^2+^. In the presence of H_2_O_2_, the CPs catalyzed the oxidation of TA, producing 2-hydroxyterephthalic acid (TAOH), which emitted blue fluorescence at 426 nm, while the fluorescence of Tb^3+^ at 546 nm was quenched. This enabled the development of a ratiometric fluorescence method where the ratio between the fluorescence intensities at 426 nm (from TAOH) and 546 nm (from Tb^3+^) directly correlated with the concentration of H_2_O_2_. The sensor exhibited a detection range from 0.01 to 300 µM and a detection limit of 1.62 nM for H_2_O_2_. Furthermore, the method was extended to GSH detection, leveraging the fact that GSH could bind to Cu^2+^, inhibiting the peroxidase-like activity of the CPs and reversing the fluorescence changes. This dual-emission ratiometric platform allowed for the precise detection of both H_2_O_2_ and GSH with high sensitivity. The study highlighted the simplicity and versatility of the ratiometric luminescence method, demonstrating that the Tb/Cu-GDP/TA CPs probe could be used for the real-time monitoring of H_2_O_2_ and GSH in both biological samples and environmental monitoring.

#### 3.2.2. Glucose

Glucose is an important monosaccharide that is widely present in living organisms and serves as one of the main sources of energy for cells. It is converted into energy through the process of glycolysis and plays a key role in the body’s metabolism, particularly in maintaining stable blood sugar levels. In the medical field, monitoring glucose is crucial for managing diabetes. To improve the accuracy and sensitivity of glucose detection, modern research focuses on developing efficient sensors.

Liu et al. developed a novel multifunctional composite based on lanthanide/nucleotide coordination polymers for ratiometric sensing, specifically targeting glucose as well as H_2_O_2_ (Figure 5) [93]. The composite was fabricated by self-assembling Tb^3+^, AMP, glucose oxidase (GOx), and carbon dots (CDs), resulting in the formation of GOx&CDs@AMP/Tb. The lanthanide/nucleotide CPs served as a robust platform for co-encapsulating enzymes and nanoparticles, enhancing the catalytic activity of GOx, which was nearly doubled compared to free GOx. Moreover, the composite demonstrated increased stability with minimal leakage of GOx. A dual-emissive GOx&CDs@AMP/Tb-CPBA composite was then created by incorporating carboxyphenylboronic acid (CPBA), which acted as a sensitizer for Tb^3+^ fluorescence and specifically recognized H_2_O_2_. Upon adding glucose, GOx catalyzed the conversion of glucose to H_2_O_2_, which induced the deborronation of CPBA and the subsequent quenching of Tb^3+^ fluorescence. This was coupled with a non-interfering fluorescence signal from the CDs at 445 nm, enabling ratiometric detection by measuring the fluorescence intensity ratio between CDs at 445 nm and Tb^3+^ at 544 nm. The ratiometric fluorescence method provided ultrahigh sensitivity with a detection limit as low as 80 nM for glucose, allowing easy identification by the naked eye under UV light. The sensor proved to be highly selective and resistant to environmental fluctuations, making it suitable for practical applications in biosensing. This work offered a new insight into the development of lanthanide/nucleotide CPs as enzyme-encapsulating platforms, demonstrating their potential in multifunctional composite creation for sensing applications, particularly for ratiometric glucose and H_2_O_2_ detection.

Researchers have also developed other ratiometric glucose sensing systems using lanthanide coordination polymers to encapsulate GOx. Li et al. developed a dual-emission lanthanide coordination polymer for the ratiometric detection of H_2_O_2_ and glucose [94]. Tb-ATA/GDP CPs were prepared by self-assembling guanine diphosphate (GDP), 2-aminoterephthalic acid (ATA), and Tb^3+^. GDP served as the antenna ligand, sensitizing the luminescence of Tb^3+^, while ATA acted as an auxiliary ligand, providing a reference fluorescence signal for the ratiometric assay. Under single-wavelength excitation, the probe emitted the blue fluorescence of ATA at 430 nm and the green fluorescence of Tb^3+^ at 544 nm. Upon the addition of H_2_O_2_, the fluorescence of Tb^3+^ was quenched, while ATA’s fluorescence at 430 nm increased, leading to a ratiometric fluorescence change for the accurate detection of H_2_O_2_. Subsequently, researchers utilized the adaptive nature of Tb-ATA/GDP chlorides to incorporate GOx into the system. This allowed the probe to detect glucose through the H_2_O_2_ generated by the glucose oxidation reaction catalyzed by GOx. This method provides a simple, highly sensitive, and highly selective ratiometric fluorescence detection strategy for glucose. Huang and Wu et al. developed a ratiometric fluorescence glucose sensor based on heteronuclear lanthanide coordination polymer nanoparticles [95]. The composite, named RhB&GOx@AMP-Tb/Ce, was prepared by self-assembling AMP, Ce^3+^, and Tb^3+^, and encapsulating rhodamine B (RhB) and glucose oxidase (GOx) as guest molecules within the coordination polymer. Due to the luminescent properties of lanthanide ions, and the catalytic action of GOx in glucose oxidation to produce H_2_O_2_, the composite showed enhanced catalytic activity and stability compared to free GOx, thanks to the confinement effect of the LnCPs. The fluorescence of Tb^3+^ was sensitized through energy transfer from Ce^3+^ to Tb^3+^, and subsequently from Tb^3+^ to RhB, forming a dual-emission system with fluorescence peaks at 549 nm (for Tb^3+^) and 575 nm (for RhB). Upon adding glucose, the H_2_O_2_ generated by GOx catalysis oxidized Ce^3+^ to Ce^4+^, disrupting the energy transfer process and causing a ratiometric fluorescence change. Specifically, the fluorescence of Tb^3+^ decreased, while the fluorescence of RhB increased, enabling the quantitative detection of glucose. Based on the H_2_O_2_-induced conversion of Ce^3+^ to Ce^4+^ and the associated fluorescence change, He et al. reported a ratiometric fluorescence probe based on the Ce(III)-ATP-fluorescein coordination polymer, which was applied in an enzyme-linked immunosorbent assay (ELISA) [96].

#### 3.2.3. Hydrogen Sulfide

Hydrogen sulfide (H_2_S) is a colorless gas with a strong rotten egg smell, and is widely found in nature and in biological systems [97,98,99,100,101]. As a gas signaling molecule, H_2_S plays a crucial role in the human body, particularly in regulating vascular tension, neurotransmission, and immune responses. It causes vascular smooth muscle cells to regulate vasodilation, thus playing a role in blood pressure regulation; simultaneously, it serves as a neurotransmitter involved in neuroregulation and cell protection. The concentration of H_2_S is closely related to the development of cardiovascular diseases, neurodegenerative diseases, and cancer, making its detection of great significance in disease diagnosis and clinical monitoring.

Tan et al. developed a ratiometric sensor for H_2_S detection based on the copper (Cu^2+^)-mediated fluorescence of lanthanide coordination polymers doped with carbon dots (CDs) [102]. The sensor, labeled as CDs@ZIF-8@GMP/Tb, was fabricated by self-assembling terbium ions (Tb^3+^) and guanosine monophosphate (GMP) onto the surface of a carbon dots-loaded zeolitic imidazolate framework (ZIF-8). In this sensor, CDs served as the reference signal due to their stable fluorescence, while Tb^3+^ in the LnCPs acted as the response signal. The system exhibited dual emissions: blue fluorescence from CDs at 430 nm and green fluorescence from Tb^3+^ at 544 nm. Upon adding Cu^2+^, the Tb^3+^ fluorescence was quenched, and it was recovered when H_2_S was added, demonstrating a typical ON–OFF–ON fluorescence behavior. By measuring the fluorescence intensity ratio at 430 nm and 544 nm (I_430_/I_544_), the highly sensitive and selective detection of H_2_S was achieved with a detection limit of 150 nM, which could be visually observed at concentrations as low as 3 μM. The sensor exhibited excellent selectivity for H_2_S against other anions and biological species, making it a valuable tool for environmental monitoring and biological applications. This study demonstrated the potential of integrating LnCPs and carbon dots to develop advanced ratiometric fluorescent sensors, offering high sensitivity, simplicity, and resistance to environmental interference.

Zeng et al. developed a ratiometric fluorescence probe for sulfide ions (S^2−^, a form of H_2_S that predominantly exists as HS^−^ under physiological conditions) based on lanthanide coordination polymers doped with carbon dots (CDs) (Figure 6) [103]. The sensor, called Coumarin@GDP-Tb, was synthesized by self-assembling guanosine diphosphate (GDP) with terbium ions (Tb^3+^) and carboxy-coumarin. Under a single excitation wavelength of 290 nm, the probe displayed dual emissions: blue fluorescence from coumarin at 440 nm and green fluorescence from Tb^3+^ at 545 nm. The ratiometric fluorescence sensing mechanism is based on the selective quenching of Tb^3+^ fluorescence at 545 nm by Fe^2+^ ions and fluorescence recovery in the presence of S^2−^, while the coumarin fluorescence at 440 nm remained unaffected by Fe^2+^ or S^2−^. Based on this, a ratiometric fluorescence assay for S^2−^ ions was developed by using the fluorescence intensity ratio (F_545_/F_440_) as the analytical signal.

#### 3.2.4. Dopamine

Dopamine (DA) is a neurotransmitter that plays a critical role in various physiological processes, especially in the brain [104,105,106]. It is involved in the regulation of mood, behavior, movement, and cognition. Dopamine is synthesized in the brain from the amino acid tyrosine and plays a key role in the function of multiple brain regions, particularly those responsible for pleasure, reward, and motor control. Abnormal dopamine levels are closely associated with various neurological disorders, such as Parkinson’s disease, Alzheimer’s disease, schizophrenia, and Huntington’s disease. Due to its role in regulating mood and cognition, dopamine is also important in psychiatric conditions. Given its crucial role in health and disease, the sensitive and selective detection of dopamine levels is vital for diagnosing and monitoring these neurological and psychiatric disorders. Li et al. developed a ratiometric fluorescence sensor for the sensitive detection of dopamine based on copper-doped lanthanide coordination polymers [107]. The sensor, labeled as Tb/Cu-GMP/ATA CPs, was designed by self-assembling GMP, 2-aminoterephthalic acid (ATA), Cu^2+^, and Tb^3+^. The GMP molecule acted as an antenna ligand to sensitize the fluorescence of Tb^3+^, while ATA provided a reference fluorescent signal. Under single-wavelength excitation, the system exhibited dual emissions: green fluorescence from Tb^3+^ at 544 nm and blue fluorescence from ATA at 430 nm. Upon the addition of DA, the system catalyzed the oxidation of DA into polydopamine (PDA), which quenched the green fluorescence from Tb^3+^ due to an internal filtration effect. This fluorescence change allowed for the construction of a ratiometric sensor for dopamine detection, with the fluorescence intensity ratio (I_544_/I_430_) being linearly correlated with the DA concentration in the range of 1 to 400 µM. The detection limit of the sensor was determined to be 0.44 µM. The ratiometric fluorescence detection method enhanced the accuracy and stability of DA measurements, overcoming the challenges posed by environmental fluctuations and instrument interference. The sensor was successfully tested in human serum samples, showing good recovery and results highly consistent with those obtained using high-performance liquid chromatography (HPLC).

#### 3.2.5. Histamine

Histamine is a biogenic amine that is widely present in the human body and other animals, primarily acting as a neurotransmitter and immune modulator [108,109]. It plays a key role in allergic reactions, gastric acid secretion, and immune system regulation. Histamine is produced from the amino acid histidine through the action of histidine decarboxylase. In the body, histamine is mainly stored in mast cells and basophils. An excess of histamine can trigger a range of physiological responses, such as allergic reactions, headaches, vasodilation, hypotension, and skin irritation. Especially during food spoilage, microorganisms break down proteins to produce excessive histamine, and consuming foods with a high histamine content can lead to histamine poisoning, causing allergic reactions. Therefore, detecting the histamine levels in food is crucial for ensuring food safety. Chen and Wang et al. presented a ratiometric fluorescence probe based on hydrogen-bonded organic frameworks (HOFs) and Eu^3+^ for histamine detection, enabling the real-time monitoring of seafood freshness [110]. The probe was self-assembled through hydrogen bonding, combining amine-reactive fluorescein isothiocyanate (5-FITC) and Eu^3+^. Amine-reactive gases formed stable complexes with the HOF through hydrogen bonding, causing the probe to emit dual-characteristic fluorescence at different wavelengths (525 nm and 616 nm). As the histamine concentration changed, the fluorescence intensities at 525 nm (green) and 616 nm (red) exhibited an inverse correlation, greatly enhancing the sensitivity and accuracy of detection. This probe demonstrated a sensitive red–green fluorescence change, which was used for the on-site monitoring of seafood freshness. By applying the probe to composite films and using a smartphone combined with a 254 nm UV lamp, rapid and intuitive on-site monitoring was achievable. The experimental results showed that this probe could detect the histamine content in seafood, providing clear visual signals with significant color changes, making it highly valuable for applications. The red–green fluorescence changes and self-calibration properties made it an ideal tool for food quality assessment and food safety detection.

#### 3.2.6. Tryptophan

Tryptophan (Trp) is an essential amino acid that is widely present in the human body and other organisms [111,112,113]. It is an important component of proteins and plays a critical role in human metabolism, growth, and development. Tryptophan is not only a precursor of the neurotransmitter serotonin, but also participates in regulating mood, sleep, appetite, and pain. A deficiency in tryptophan is associated with various emotional problems such as irritability, anxiety, and insomnia, and a lack of tryptophan can lead to neurological disorders such as depression. Additionally, tryptophan and its metabolites, such as 5-hydroxyindoleacetic acid (5-HIAA), are commonly used as important biomarkers in clinical settings to monitor the occurrence and progression of diseases, especially carcinoid tumors.

Yan et al. proposed a novel ternary hybrid material combining Eu(III)-functionalized hydrogen-bonded organic frameworks (HOFs) and covalent organic frameworks (COFs), which was synthesized by post-synthetic modification to obtain the Eu@HOF/COF composite [114]. This composite, functioning as a ratiometric fluorescence sensor, is capable of quantifying Trp and its metabolite 5-hydroxyindoleacetic acid (5-HIAA), and can also be applied to latent fingerprint (LFP) identification. The Eu@HOF/COF composite exhibits dual emission at 510 nm and 616 nm under single excitation, corresponding to COF and Eu(III), respectively. The post-modification with Eu(III) induces structural changes in the organic linkers of HOF, enhancing the luminescence of Eu(III). The presence of tryptophan interferes with the fluorescence behavior of the Eu@HOF/COF composite. Tryptophan interacts with the ligands and metal ions in the Eu@HOF/COF composite, affecting the fluorescence emission of Eu(III). As the concentration of tryptophan increases, the red fluorescence of Eu(III) is quenched, while the blue fluorescence of COF remains unchanged or undergoes slight variation. The fluorescence intensity ratio (i.e., F_616_/F_510_) shows a linear relationship with the tryptophan concentration. The Eu@HOF/COF sensor enables the rapid and sensitive detection of the tryptophan concentration in complex biological samples such as serum and urine.

#### 3.2.7. Dipicolinic Acid

Dipicolinic acid (DPA) is an important organic compound widely involved in biomedical and environmental monitoring fields, especially in the detection of bacterial spores [115,116,117,118]. It is a nitrogen-containing aromatic acid with strong stability that can form complexes with calcium ions. During the formation of bacterial spores, DPA binds with calcium ions to help the bacteria withstand extreme environmental conditions, such as high temperatures and chemical toxicity. This makes DPA a crucial biomarker in spore-forming bacteria, such as Bacillus anthracis spores. DPA is widely applied in biosensors, especially for detecting anthrax spores. By utilizing the fluorescent properties of DPA, researchers have been able to design highly sensitive sensors for early diagnosis, ensuring biological safety and environmental monitoring. Despite its importance in biosafety, challenges remain in improving its detection sensitivity and reducing costs, especially in complex environments. With technological advancements, DPA detection methods will play an increasingly important role in disease control, environmental monitoring, and public safety.

Zhang et al. developed a dual lanthanide-doped time-resolved ratiometric fluorescent probe for detecting DPA [119]. The probe combines terbium (Tb^3+^) and europium (Eu^3+^)-doped composite materials, utilizing time-resolved fluorescence and ratiometric fluorescence detection techniques to achieve high sensitivity, selectivity, and accuracy for DPA detection. First, the researchers synthesized Tb/DPA@SiO_2_ composites using a reverse microemulsion method, enhancing their stability by introducing a silica carrier. The terbium ions (Tb^3+^) were incorporated within the silica (SiO_2_), providing a stable green fluorescence reference signal, while Eu/GMP complexes formed on the surface of SiO_2_ by coordination interactions. In the presence of DPA, the fluorescence of the Eu/GMP part increased, while the green fluorescence of Tb/DPA@SiO_2_ remained unchanged. By monitoring the fluorescence intensity ratio between the red and green emissions, the DPA concentrations could be effectively quantified. Additionally, the probe utilized time-resolved fluorescence technology to separate long-lived Eu^3+^ fluorescence signals from short-lived background fluorescence, significantly reducing the autofluorescence interference caused by biological samples. The probe demonstrated excellent sensitivity, accurately measuring DPA concentrations in the range of 1 to 400 µM, with a detection limit of 0.44 µM. The author also developed a paper-based sensor by embedding the Tb/DPA@SiO_2_-Eu/GMP probe into filter paper, observing a distinct fluorescence color change from green to red under UV light, enabling simple on-site detection. The sensor is stable, portable, low-cost, and disposable, making it suitable for rapid on-site testing. Coupled with a smartphone and a color-scanning app, users can perform quantitative scanometric measurements, further expanding the application of this technology, particularly in low-cost and rapid detection scenarios. Recently, Gu et al. encapsulated 8-hydroxypyrene-1,3,6-trisulfonic acid trisodium salt (HPTS) in silica nanoparticles and combined it with a europium-based guanine mononucleotide (Eu/GMP) coordination polymer to develop another ratiometric fluorescence sensing method for DPA detection [120].

Tang et al. developed a novel core–shell lanthanide-functionalized micelle nanoprobe for the rapid, sensitive, and selective detection of the anthrax biomarker DPA [121]. The probe was constructed using a “one-pot” self-assembly method, where an amphiphilic ligand containing a β-diketone derivative was used to immobilize terbium ions (Tb^3+^) through coordination interactions. Additionally, a fluorophore emitting blue fluorescence, sodium 6-(dimethylamino)naphthalene-2-sulfonate (FR), was incorporated into the micelle as a reference signal. Upon the addition of DPA, the fluorescence intensity of the fluorophore remained essentially unchanged, while the Tb^3+^ fluorescence increased due to energy transfer from the DPA chromophore. This ratiometric fluorescence response allowed for the accurate quantification of DPA levels. The probe demonstrated high sensitivity and selectivity for DPA, with a linear relationship observed between the fluorescence ratio and DPA concentration in the range of 0 to 7.0 µM. The probe also exhibited excellent selectivity for DPA over other aromatic ligands, making it particularly suitable for anthrax spore detection. The study also introduced a visual detection method using a paper-based sensor embedded with the probe, where the fluorescence color changed from green to red under UV light, making it visible to the naked eye. Additionally, the results could be quantitatively scanned and analyzed using a smartphone with a color-scanning app, providing a portable, cost-effective, and easy-to-use solution for on-site detection.

Deng and Zhou et al. developed a novel dual ratiometric fluorescence probe for detecting DPA based on the stimulus response of tetra(4-sulfophenyl)ethene (TPE-TS)-functionalized infinite coordination polymer (ICP) nanoparticles (Figure 7) [121]. The probe was constructed using a one-pot self-assembly method, in which the fluorescent guest TPE-TS was encapsulated within the Eu/GMP ICP host. The resulting TPE-TS@Eu/GMP ICP nanoparticles exhibited dual emission, which was due to the aggregation-induced emission (AIE) effect of TPE-TS and the sensitized fluorescence from Eu^3+^ ions. Upon the addition of DPA, the structure of the complex was disrupted, causing the release of TPE-TS and resulting in a shift in fluorescence from blue to red, enabling the highly sensitive and selective detection of DPA. The fluorescence mechanism was based on a triple response: the monomer emission (ME) of TPE-TS at 390 nm, the AIE at 455 nm, and the Eu^3+^-sensitized fluorescence at 592, 615, 652, and 693 nm. The addition of DPA enhanced the red fluorescence of Eu^3+^, disrupted the host–guest structure, and caused a decrease in AIE, leading to a dual ratiometric fluorescence response. This response enabled the quantitative detection of DPA levels in the range of 0 to 7.0 µM with high sensitivity and selectivity, making it ideal for anthrax spore detection. Furthermore, a multichannel signal detection system was established using coffee ring-based nanochromatography. The response of the ICP nanoparticles to DPA was visualized as distinct deposition patterns on test paper, which were then analyzed through an image recognition application on a smartphone. This innovative system provided a portable, low-cost, and high-throughput method for the real-time detection of DPA, offering an effective solution for anthrax biomarker monitoring.

### 3.3. Small Molecule Drugs

#### 3.3.1. Ciprofloxacin and Pefloxacin

Ciprofloxacin (CIP) and pefloxacin (PFLX) are both fluoroquinolone antibiotics, and are widely used for the treatment of bacterial infections, particularly those caused by Gram-negative bacteria [122,123,124]. These two drugs share similar chemical structures and mechanisms of action. Both ciprofloxacin and pefloxacin inhibit bacterial DNA replication by targeting bacterial DNA gyrase and topoisomerase IV, enzymes essential for DNA replication and transcription. By blocking these enzymes, they prevent the bacteria from reproducing, thus inhibiting bacterial growth. Ciprofloxacin is a second-generation fluoroquinolone known for its effectiveness against a broad range of Gram-negative bacteria like Escherichia coli and Salmonella, as well as some Gram-positive bacteria. It is commonly used for treating urinary tract infections, respiratory infections, skin infections, and more. Pefloxacin, on the other hand, is used primarily for respiratory, urinary, and skin infections. Although effective, it is less commonly used than ciprofloxacin due to its higher side effect profile. Both drugs are crucial in clinical settings and veterinary medicine. However, the improper or overuse of either ciprofloxacin or pefloxacin can contribute to the development of antibiotic-resistant bacteria, posing a significant challenge to global health. Therefore, it is important to monitor their residues in clinical samples, food, and the environment to minimize the spread of resistance.

Li and Ye developed a ratiometric fluorescence sensor for CIP detection using terbium-based coordination polymers [125]. The sensor was constructed through a simple self-assembly process involving Tb^3+^, AMP, and luminol as dual ligands. AMP and luminol formed a coordination polymer, where luminol provided a stable fluorescence reference signal, and terbium ions served as the responsive signal for CIP detection. The detection principle relied on the ability of CIP to sensitize the luminescence of terbium ions. Upon the introduction of CIP, it coordinated with Tb^3+^ ions, enhancing their green fluorescence through an energy transfer mechanism, while the fluorescence intensity of luminol remained unchanged. This dual-emission system allowed for the calculation of the fluorescence intensity ratio between Tb^3+^ and luminol, which showed a linear relationship with the CIP concentration. The sensor exhibited high sensitivity with a detection limit of 2 nM and a dynamic range from 5 nM to 2.5 µM. It also demonstrated excellent selectivity for CIP, providing reliable results even in complex biological samples such as human serum. This study represented the first application of a dual-ligand coordination polymer for ratiometric CIP detection, offering a stable and accurate sensing platform with potential applications in environmental and biological monitoring. Recently, Liu, Hao, and Zhou et al. presented a sensitive and selective ratiometric fluorescence nanoprobe, GMP-Tb-BDC, for CIP detection [126]. By utilizing terbium ions sensitized by CIP to enhance fluorescence and by employing BDC as a stable internal reference, the probe was successfully applied to real samples with on-site detection through mobile phone analysis.

Jia and Xu et al. presented a ratiometric fluorescent nanoprobe for the intelligent detection of PFLX, addressing the need for a rapid, sensitive, and visual detection method for PFLX residues in the environment [127]. The probe was composed of bovine serum albumin-coated gold nanoclusters (BSA-AuNCs) and a guanine nucleotide-terbium ion (GMP-Tb) complex. The AuNCs provided stable fluorescence properties, while the GMP-Tb complex was used for its unique luminescent behavior, including high photochemical stability and sharp emission peaks. The detection principle relied on the ratiometric fluorescence shift from red to green as the concentration of PFLX increased, due to the sensitization effect between PFLX and the Tb^3+^ ion. The probe demonstrated a wide detection range (0.01 μM to 60 μM) and an ultra-low detection limit (5.37 nM), making it highly sensitive. A paper-based sensor was designed for the on-site, visual detection of PFLX, which could be analyzed using a smartphone app with color recognition. To further enhance the detection process, a web-based system was developed, incorporating a YOLO v5 target detection algorithm and machine learning (ML) regression algorithms for the quantitative analysis of PFLX concentrations in real environmental samples. The web system allowed for the batch detection of PFLX, greatly improving the detection speed. This study highlighted the successful integration of ratiometric fluorescence sensing with machine learning, offering a simple, efficient, and portable method for the environmental and food safety monitoring of PFLX residues.

#### 3.3.2. Tetracycline and Oxytetracycline

Tetracycline (Tc) and Oxytetracycline (OTC) are both broad-spectrum antibiotics belonging to the tetracycline class, and are widely used to treat bacterial infections [128,129,130,131]. Their mechanism of action primarily involves binding to the 30S ribosomal subunit of bacteria, preventing the attachment of aminoacyl-tRNA, thereby inhibiting protein synthesis and bacterial growth. Tetracycline is commonly used to treat pneumonia, urinary tract infections, typhoid, and acne, while Oxytetracycline is more frequently applied in veterinary medicine and agricultural antibiotic use. Despite their important role in antimicrobial therapy, the prolonged misuse or overuse of these antibiotics can lead to bacterial resistance. Therefore, the rational use of these antibiotics, along with the development of highly sensitive detection methods to monitor their residues in the environment and food, is crucial for ensuring public health and environmental safety.

Zhang et al. developed a programmable printed paper-based device for the ratiometric fluorescence detection of Tc in various natural samples [132]. The device uses composite probe ink, consisting of MoS_2_ nanoparticles (NPs) and GMP/Eu-Cit (an Eu-based coordination polymer). The detection principle relies on two distinct fluorescence signals: MoS_2_ NPs quench fluorescence at 430 nm when interacting with Tc, while GMP/Eu-Cit emits strong red fluorescence at 617 nm. Ratiometric analysis depends on the alternating intensities of these signals in response to different TC concentrations. TC coordinates with Eu^3+^ ions in GMP/Eu-Cit, transferring energy to enhance the red fluorescence, while MoS_2_ NPs experience fluorescence quenching due to photoinduced electron transfer and the inner filter effect. The device uses programmable printing to optimize the distribution of probes, enhancing sensor performance. A two-stage printing process creates transitional devices by gradually altering the number of probes, allowing the device to automatically adjust and optimize probe distribution for efficient Tc detection in complex backgrounds. By integrating with a smartphone app and a 3D-printed measurement chamber, the sensor’s capability is enhanced. The device successfully detected Tc over a wide concentration range (12.7 nM to 80 μM), with excellent selectivity and stability, providing visible signals and smartphone-based readings. The device was tested on various natural samples, including soil, river water, milk, and serum, with satisfactory results consistent with HPLC-MS findings. The programmable printing feature effectively improved device performance, making it a useful tool for on-site TC monitoring in complex environmental samples. Tong et al. reported a ratiometric fluorescence sensor based on lanthanide coordination polymer nanoparticles (luminol-Eu^3+^-GMP CPNPs) for the real-time and visual detection of Tc [133]. Luminol-Eu^3+^-GMP CPNPs were synthesized by self-assembly at room temperature, consisting of Eu^3+^, luminol, and guanosine monophosphate. Initially, the probe emitted only blue fluorescence, with no red characteristic fluorescence from Eu^3+^. In the presence of TC, TC triggered the antenna effect (AE), turning on the red fluorescence of Eu^3+^, while simultaneously quenching the blue fluorescence of luminol-Eu^3+^-GMP CPNPs through the inner filter effect (IFE). The addition of sodium citrate (Cit) further enhanced the red fluorescence of Eu^3+^. This sensor utilized dual fluorescence responses induced by TC to achieve ratiometric fluorescence detection, with a low detection limit (3.4 nM), far below the maximum residue limits (MRLs) of tetracycline in milk (100 mg/kg in China and 225 nM in the EU). Additionally, the sensor exhibited a rapid response, reaching fluorescence equilibrium in just 30 S, making it suitable for real-time detection. The study also demonstrated the development of point-of-care testing (POCT) devices combined with smartphones and test paper for the real-time and semi-quantitative detection of tetracycline. The sensor was successfully applied to tetracycline detection in honey, milk, lake water, and tap water samples.

Tian et al. developed a novel ratiometric fluorescence probe based on carbon dots-doped europium(III) coordination polymers (CD@AMP/Eu NCPs) for the visual and quantitative detection of OTC [134]. The probe was formed by the self-assembly of Eu^3+^ and AMP on the surface of carbon dots (CDs), which possess abundant hydroxyl and carbonyl groups. Under 310 nm excitation, the probe exhibited dual-emission fluorescence: Eu^3+^ emitted strong pink light at 615 nm, and the CDs emitted blue light at 430 nm. Upon exposure to OTC, the fluorescence of Eu^3+^ was enhanced while the fluorescence of CDs was quenched, allowing for OTC detection. For practical application, the NCP was immobilized on filter paper to create paper-based test strips for quantitative OTC analysis. Using a digital camera and an app-based color detector, the paper-based strips displayed a wide linear range of 1 to 100 μM with a detection limit of 0.5 μM, enabling the rapid and intuitive monitoring of OTC in milk samples. Deng et al. developed a pH-regulated sensor array based on 1,1,2,2-tetra(4-carboxylphenyl)-ethylene (H_4_TCPE), which was incorporated into europium(III) coordination polymers (Eu/AMP ICP) for the systematic analysis of environmental antibiotics (Figure 8) [135]. This sensor array, referred to as the “AIE@Ln/ICP lab-on-a-chip” system, was designed to simultaneously identify and sense multiple antibiotics in environmental samples. The probe was composed of H_4_TCPE as the guest and Eu/AMP ICP as the host, where the unique aggregation-induced emission (AIE) properties of H_4_TCPE and the time-resolved fluorescence (TRF) of Eu^3+^ were utilized as key sensing elements. The sensing mechanism involved the antenna effect of Eu/AMP ICP and the reductive photoinduced electron transfer (PET) between H_4_TCPE and the antibiotics. When exposed to various antibiotics, such as flumequine, oxytetracycline, and sulfadiazine, the probe showed distinct fluorescence responses that were pH-regulated, enabling systematic antibiotic analysis. Principal component analysis (PCA) was employed to process the fluorescence data, successfully distinguishing antibiotics with different structural characteristics, including the tetracycline subclass (oxytetracycline, tetracycline, and doxycycline). The sensor array exhibited significant anti-aggregation-caused quenching (ACQ) effects, producing clear fluorescence color changes when applied to test paper. These changes were easily recognized using a smartphone, facilitating antibiotic identification based on fluorescence fingerprinting on test paper. The sensor array demonstrated high sensitivity, reliability, and a rapid response, with great potential for field environmental antibiotic analysis, particularly in resource-limited areas. This study emphasizes the innovative use of the H_4_TCPE@Eu/AMP ICP sensor array, addressing challenges such as aggregation-induced quenching and the need for reliable multiple antibiotic detection, showcasing its potential for widespread environmental and on-site applications.

### 3.4. Biomacromolecules

#### 3.4.1. Alkaline Phosphatase

Alkaline phosphatase (ALP) is an essential enzyme involved in various physiological processes, including bone mineralization, liver function, and detoxification [136,137,138]. It catalyzes the hydrolysis of phosphate esters, producing inorganic phosphate and alcohol, primarily in an alkaline environment. ALP exists in different isoenzymes found in the liver, bones, kidneys, and intestines, with each isoenzyme playing a distinct role in specific tissues. In bones, ALP contributes to the release of phosphate for the formation of hydroxyapatite, crucial for bone health. In the liver, it helps in bile secretion and detoxification. Clinically, ALP is a key biomarker for diagnosing conditions like liver diseases, bone disorders, and bile duct obstructions. Modern detection methods, particularly ratiometric fluorescence assays, have enabled the highly sensitive and specific monitoring of ALP activity, making it a valuable tool in medical diagnostics, environmental testing, and food safety.

ALP can hydrolyze GMP, leading to structural changes in the lanthanide coordination polymers composed of GMP, which affect the luminescent properties of the lanthanide center, particularly the fluorescence of Tb sensitized by GMP. This induces fluorescence quenching, which, when combined with changes in the fluorescence of another luminescent center, enables the detection of ALP. Based on this principle, a series of ratiometric fluorescence detection systems have been developed. Mao et al. developed a novel ratiometric fluorescent method for real-time ALP activity assay using stimulus-responsive infinite coordination polymer (ICP) nanoparticles as the probe [139]. The ICP nanoparticles consisted of two components: a supramolecular ICP network formed by GMP as the ligand and Tb^3+^ as the central metal ion, and a fluorescent dye, 7-amino-4-methyl coumarin (coumarin), encapsulated within the ICP network. In the absence of ALP, the coumarin@Tb-GMP nanoparticles exhibited stable green fluorescence at 552 nm, while the coumarin dye emitted weak fluorescence at 450 nm. Upon the addition of ALP, the enzyme catalyzed the cleavage of the phosphate group in the GMP ligand, leading to the disruption of the ICP network. This process caused the green fluorescence from Tb^3+^ to decrease, while the fluorescence of the coumarin released at 450 nm became stronger. The ratio of the fluorescence intensities (F_552_/F_450_) provided a reliable measure of ALP activity. The assay showed a linear response for ALP activity within the range of 0.025 U∙mL^−1^ to 0.2 U∙mL^−1^, with a detection limit as low as 0.010 U∙mL^−1^. This method proved to be highly sensitive and was further applied for the evaluation of ALP inhibitors. The study demonstrated that the ratiometric method overcame the limitations of previous fluorescent assays by minimizing the impact of environmental fluctuations, such as light intensity, pH, and polarity. This approach provided enhanced sensitivity and stability, making it more suitable for practical applications. Based on similar principles, researchers have reported various ratiometric luminescent systems based on lanthanide coordination polymers for ALP activity detection by selecting different second luminescent ligands, such as Ce [140], Eu [141], Cu nanocrystals (NCs) [142], ZIF-8 [143], UiO-66-NH_2_ [144], and carbon dots (CDs) [145].

Wu and Tong et al. developed a ratiometric fluorescent probe based on the aggregation-enhanced antenna effect (AEE) for the detection of endogenous ALP activity (Figure 9) [146]. In this study, four gelatinous Ln^3+^ (Eu, Tb, Sm, Dy) coordination polymers (Ln-CPs) were synthesized by self-assembling CIP, AMP, and Ln^3+^ ions in aqueous medium. With an increase in the Ln-CP concentration, the characteristic fluorescence of Ln^3+^ was significantly enhanced, while the fluorescence of CIP decreased notably. This phenomenon was attributed to the large aggregates formed during the self-assembly process, which effectively restricted the intramolecular rotations of CIP molecules, thereby enhancing the antenna effect (AE). Specifically, Eu-CP exhibited a rice-like morphology in a highly aggregated state and demonstrated selective responsiveness to ALP. Upon interaction with ALP, the fluorescence of Eu^3+^ decreased significantly, while the fluorescence of CIP gradually recovered, accompanied by a morphological transition from a rice-like to a flower-like polymer. This process enabled the ratiometric fluorescence detection of ALP activity within the range of 0.1–6.0 U∙L^−1^, with high sensitivity. Furthermore, ALP detection in human serum and fluorescence imaging in living cells were successfully carried out with satisfactory results. The study demonstrated the use of the aggregation-enhanced antenna effect (AEE) and aggregation-induced emission (AIE) phenomena, optimizing the AE effect by modulating the aggregation state of the Ln-CP polymers. This new probe provides an efficient and easy-to-operate method for real-time ALP activity detection and overcomes the limitations of traditional methods, such as solvent effects and low sensitivity.

Liang and Qiu et al. developed a ratiometric fluorescence probe based on dual-ligand lanthanide coordination polymer nanoparticles (ThT@luminol-Eu-GMP CPNPs) for the highly sensitive detection of ALP [147]. The probe was synthesized by self-assembling Eu^3+^ with the ligands GMP and luminol, forming a dual-ligand coordination polymer. During this process, the guest dye molecule thioflavin T (ThT) was embedded, which enhanced the fluorescence performance of ThT in two ways. First, the molecular rotation of ThT was restricted due to its tight embedding within the coordination polymer, leading to an increase in ThT fluorescence. Second, Förster resonance energy transfer (FRET) between the emission spectra of luminol and the absorption spectra of ThT further facilitated the fluorescence of ThT. In the presence of ALP, the specific cleavage of the phosphate group in GMP disrupted the ThT@luminol-Eu-GMP CPNPs structure, releasing ThT, Eu^3+^, and luminol into the solution. Due to the aggregation-induced emission (AIE) effect of luminol, the fluorescence of ThT significantly decreased, while that of luminol increased. The ratiometric detection of ALP was achieved by monitoring the change in the fluorescence intensity between ThT and luminol, enabling the highly sensitive monitoring of ALP activity. Additionally, the ThT@luminol-Eu-GMP probe was also used for arsenate (As(V)) detection based on the inhibition of ALP activity by arsenate. Recently, Yu and Zhang reported a ratiometric lanthanide fluorescent probe (CIP@SiO_2_-Ce/ATP-Tris) for the sensitive detection of ALP [148]. The probe was composed of Ce^3+^ ions as the central ion, ATP as the coordination ligand, Tris as the auxiliary ligand, and CIP encapsulated in SiO_2_ nanoparticles serving as the internal reference fluorescence. The probe exhibited characteristic fluorescence at 363 nm from Ce^3+^ and at 435 nm from CIP. The detection principle was based on the specific enzymatic activity of ALP, which catalyzed the dephosphorylation of ATP. This enzymatic reaction disrupted the Ce/ATP-Tris complex, leading to fluorescence quenching at 363 nm, while the reference signal from CIP at 435 nm remained stable due to the protective effect of SiO_2_ encapsulation. The ratio of fluorescence intensities (I_435_/I_363_) was used to quantify ALP activity.

#### 3.4.2. Acetylcholinesterase

Acetylcholinesterase (AChE) is a crucial enzyme in the nervous system, and is responsible for breaking down the neurotransmitter acetylcholine, thereby regulating nerve transmission [149,150,151]. Acetylcholine plays a key role in transmitting nerve signals at synapses. AChE hydrolyzes acetylcholine into choline and acetic acid, rapidly terminating nerve impulses to ensure the precise control of nerve communication. AChE is particularly important at the neuromuscular junction, where it facilitates muscle contraction and relaxation. The dysfunction of AChE is associated with various neurological diseases, particularly Alzheimer’s disease, where its reduced activity leads to cognitive decline. AChE inhibitors, such as donepezil, are used to alleviate symptoms of Alzheimer’s disease by increasing acetylcholine levels and improving nerve function. However, AChE inhibitors can also be toxic and are commonly associated with poisoning due to exposure to organophosphate pesticides or nerve agents. In addition to its medical applications, AChE plays an important role in environmental monitoring and food safety, as its activity is often used to detect pesticide contamination, thus ensuring public health.

Deng and Zhou et al. reported a novel smartphone-based colorimetric assay with ratiometric fluorescence for the detection of AChE and organophosphorus pesticides (OPs), using graphene quantum dot (GQD)-sensitized terbium/guanine monophosphate (Tb/GMP) infinite coordination polymer (ICP) nanoparticles as the probe [152]. The assay was designed for point-of-use applications, aiming to provide an accessible, rapid diagnostic tool for AChE as a biomarker of OP poisoning. The probe consisted of GQDs, which were chosen for their abundant functional groups that could serve both as one of the signal readouts and as an antenna ligand to sensitize the green fluorescence of the Tb/GMP host. The Tb/GMP ICP nanoparticles emitted green fluorescence upon excitation at 330 nm, while the blue fluorescence of GQDs was suppressed due to confinement by the ICP host. When thiocholine (TCh), a product of acetylthiocholine hydrolysis by AChE, was present, it competed with GMP for coordination with Tb^3+^, leading to the collapse of the ICP network and the release of free GQDs. This resulted in a ratiometric fluorescent intensity change and a corresponding green-to-blue fluorescence color change, providing a dual-responsive mechanism for the colorimetric detection of AChE. Moreover, the presence of OPs inhibited AChE activity, preventing the stimulus response of the GQD@Tb/GMP ICP nanoparticles and causing the fluorescent color to change from greenish-blue to green. This color change was concentration-dependent, making it suitable for OP detection. The assay exhibited high sensitivity, good reliability, and clear color changes, making it a promising tool for on-site analysis.

#### 3.4.3. β-Amyloid Peptide

Amyloid-beta (Aβ) peptides, composed of short chains of amino acids, play a central role in the pathogenesis of Alzheimer’s disease (AD) [153,154,155,156]. These peptides are derived from a larger protein known as amyloid precursor protein (APP), which, when cleaved by specific enzymes, generates Aβ peptides typically ranging from 36 to 43 amino acids in length. In a healthy brain, Aβ peptides may have some physiological functions, such as aiding synaptic plasticity, but in Alzheimer’s disease, they accumulate and aggregate to form amyloid plaques. These plaques disrupt neuronal communication, trigger inflammation, and lead to cell death. The accumulation of Aβ is considered one of the major pathological features of AD. Therefore, detecting Aβ peptides and their aggregated forms is crucial for the early diagnosis of Alzheimer’s disease.

Deng and Zhou et al. developed a novel ratiometric fluorescent probe for detecting Aβ monomer [157]. This probe is based on carbon dot-sensitized lanthanide infinite coordination polymer (ICP) nanoparticles (CDs@Eu/GMP ICP). The CDs were encapsulated into Eu/GMP ICP nanoparticles via self-adaptive chemistry, utilizing competitive coordination interactions between the components. The CDs displayed strong blue fluorescence, while the Eu/GMP ICP nanoparticles emitted characteristic red fluorescence. The detection principle relies on competitive coordination interactions between CDs, Cu^2+^, and the Aβ monomer. In the absence of Cu^2^⁺, the CDs@Eu/GMP ICP nanoparticles exhibited fluorescence at 400 nm (from the CDs) and several peaks at 592 nm, 615 nm, 650 nm, and 694 nm (from Eu^3^⁺). Upon the addition of Cu^2^⁺, coordination with the CDs disrupted the antenna effect, causing a decrease in Eu^3^⁺ fluorescence. When the Aβ monomer was introduced, Cu^2^⁺ was replaced by Aβ, and the red fluorescence of Eu^3^⁺ was restored, while the fluorescence of the CDs remained unchanged. The stable fluorescence of the CDs served as an internal reference, allowing for self-correction and improving the reliability of the detection. The probe exhibited high sensitivity and was unaffected by interference from other species in rat brain tissues. This method successfully detected Aβ monomer in cerebrospinal fluid (CSF) and various brain regions of both normal and Alzheimer’s rats, providing a potential approach to monitoring Aβ monomer levels in biological fluids.

Based on the competitive binding of amyloid β-peptide with Cu^2+^, Xue et al. developed a ratiometric fluorescence probe for Aβ detection (Figure 10) [158]. The probe is based on a dual-emission luminescent lanthanide metal–organic coordination polymer, named luminol-Tb-GMP-Cu. It consists of Tb^3+^ as the central metal ion, Cu^2+^ as a cofactor to suppress Tb^3+^ fluorescence, GMP as a bridging ligand to enhance Tb^3+^ fluorescence, and luminol as an auxiliary ligand. The detection principle relies on the competitive coordination interaction between Aβ, Cu^2+^ and the components of the probe. In the absence of Aβ, Cu^2+^ interferes with the energy transfer from GMP to Tb^3+^, leading to the quenching of GMP-Tb fluorescence. The luminol fluorescence remains constant, providing an internal reference. After the addition of Aβ, it binds to Cu^2+^, leading to the restoration and enhancement of Tb^3+^ fluorescence and thereby increasing the emission of GMP-Tb. The ratio of luminol fluorescence (F_430_) to Tb^3+^ fluorescence (F_547_) enables the sensitive and accurate detection of Aβ.

### 3.5. Others

#### 3.5.1. Water Content

Water content detection is crucial across various industries such as food, pharmaceuticals, environmental monitoring, and manufacturing [159,160]. In the food and beverage industry, accurate moisture measurement is essential for quality control, as the water content directly impacts the texture, taste, shelf life, and microbial stability of products. Similarly, in the pharmaceutical industry, controlling the moisture content in formulations ensures the stability and efficacy of active ingredients, as excessive moisture can lead to degradation. In environmental monitoring, measuring the moisture in soil and water bodies helps assess quality and optimize agricultural practices. Furthermore, in industrial applications, precise moisture control is vital for material processing, product quality, and compliance with environmental regulations. Accurate water content detection methods ensure product integrity, safety, and regulatory compliance.

Deng and Zhou et al. proposed a ratiometric fluorescence detection method based on a dual-responsive lanthanide coordination polymer for the real-time detection of the alcohol content in alcoholic beverages [161]. The Ln-ICP used in this study consists of two components: one is a supramolecular Ln-ICP network formed by the coordination of 2,2′-thiodiacetic acid (TDA) with Eu^3+^, and the other is the fluorescent dye, coumarin 343 (C343), which serves both as a ligand and a sensitizer, embedded into the Ln-ICP network via self-adaptive chemistry. Upon excitation at 300 nm, energy transfer from C343 to Eu^3+^ enhances the red fluorescence at 617 nm from the Ln-ICP network, while suppressing the fluorescence of C343 at 495 nm. In pure ethanol, the C343@Eu-TDA dispersion is stable and well-dispersed. However, when water is added to the ethanol dispersion, the Eu-TDA network structure is disrupted, resulting in the release of C343 from the ICP network into the solution. This leads to the quenching of Eu-TDA fluorescence and the activation of C343 fluorescence, causing the fluorescence color of the dispersion to shift from red to blue, providing a new mechanism for the ratiometric fluorescence detection of alcohol content. This method allows for the direct, visible detection of the alcohol content in alcoholic beverages, with a linear range of 10% to 100% under UV light (365 nm).

Liu et al. developed a smartphone-based ratiometric fluorescence sensing platform for the sensitive detection of the water content in organic solvents [162]. The platform uses lanthanide-based infinite coordination polymers (Ce-GMP-DPA@Tb-DPA) as signal probes. Ce-GMP-DPA and Tb-DPA exhibit blue and green fluorescence emission characteristics, respectively. Upon complexation, the energy transfer between Ce-GMP-DPA and Tb-DPA results in the suppression of blue fluorescence and the formation of a green fluorescence from the Ce-GMP-DPA@Tb-DPA complex. The presence of water decomposes Tb-DPA, blocking the energy transfer from Ce to Tb, weakening the green fluorescence and restoring the blue fluorescence. By monitoring the ratio of blue-to-green fluorescence, the water content can be quantitatively detected. This ratiometric fluorescence method demonstrates a wide linear range of 0.2% to 90.0% water content in ethanol, with very low detection limits (0.16% in ethanol, 0.62% in tetrahydrofuran, and 0.0076% in acetone). The study further integrated a smartphone equipped with a Color Picker with the sensing platform for easy signal reading and analysis, enabling rapid and accurate water content detection with significant potential for on-site applications.

#### 3.5.2. pH

pH detection is essential in many fields, including healthcare, environmental monitoring, industrial applications, and scientific research [163,164,165]. In the medical field, monitoring the pH levels in body fluids like blood, urine, and saliva is crucial for diagnosing and managing conditions such as acidosis or alkalosis. In environmental monitoring, pH is key to assessing water quality and soil health, as extreme pH levels can harm ecosystems and affect plant growth. Industrial applications, including chemical manufacturing and water treatment, rely on precise pH control to ensure product quality, stability, and compliance with regulations. In the food and beverage industry, pH affects taste, texture, and preservation, making it essential for maintaining product safety and consistency. Additionally, in research, accurate pH measurement is vital for studying chemical reactions, optimizing experimental conditions, and developing advanced sensors. Therefore, pH detection is fundamental for ensuring health, safety, and efficiency in various sectors, with important roles in both everyday applications and cutting-edge scientific research.

Wang et al. developed a ratiometric fluorescence probe based on lanthanide–CIP complexes for efficient pH fluctuation detection [166]. The study used Eu^3+^ and Tb^3+^ as core components of the probe, with CIP coordinating with lanthanide ions to form two distinct metal complexes. The coordination groups in CIP’s molecular structure effectively bind to lanthanide ions, allowing the probe to maintain stable fluorescence characteristics in aqueous solutions. The probe utilizes the unique photoluminescent properties of lanthanide metals, particularly their narrow emission peaks and long excited-state lifetimes. As the pH of the solution changes, the probe’s spectral properties show significant variation. Specifically, under acidic conditions (pH = 3), the Eu^3^⁺ complex emits blue fluorescence, while the Tb^3^⁺ complex shows a clear on–off fluorescence change between pH 5 and 6. As the pH increases from 6 to 10, the red emission from the Eu^3^⁺ complex is significantly enhanced. These two metal complexes provide blue, green, and red fluorescence responses, enabling the visual monitoring and rapid screening of pH fluctuations. The probe exhibited a strong fluorescence response across a pH range of 3 to 10, with the Eu^3+^ complex showing intense blue emission at pH 3, and the Tb^3+^ complex exhibiting noticeable fluorescence changes from pH 5 to 6. When the pH increased from 6 to 10, the red emission of Eu^3+^ was enhanced significantly. Through the fluorescence changes of these two complexes, the researchers achieved precise pH detection, allowing the real-time monitoring of pH fluctuations with different fluorescence colors. The study further demonstrated the application of these molecular probes in HeLa cells using fluorescence microscopy, showing their ability to effectively detect pH fluctuations in living organisms, with low cytotoxicity and excellent photoluminescent properties.

#### 3.5.3. Temperature

Temperature sensing plays a critical role in various fields, including industrial processes, healthcare, environmental monitoring, food safety, and scientific research [167,168,169]. In industries such as chemical manufacturing and pharmaceuticals, temperature control is crucial for ensuring optimal conditions for chemical reactions and product quality. In healthcare, accurate body temperature measurements are essential for diagnosing and monitoring diseases such as infections, inflammatory disorders, and metabolic disturbances. Temperature is also important in environmental monitoring, where it is used to track climate change and assess ecosystem health. In the food industry, maintaining proper temperature during storage and transportation is key to preventing spoilage and ensuring food safety. Additionally, precise temperature control is indispensable for reproducibility and accuracy in scientific research and laboratory experiments.

Sobrinho and Sigoli et al. developed a water-soluble ratiometric nanothermometer composed of poly(N-isopropylacrylamide) (pNIPAM) nanoparticles grafted with lanthanide complexes, particularly trivalent lanthanide ions such as Tb^3+^ and Eu^3+^ (Figure 11) [170]. The system was designed to provide high-sensitivity and accurate temperature measurements in the submicron range. The key feature of this nanothermometer is the incorporation of a chelator monomer, a derivative of dipicolinic acid, which was synthesized through free radical emulsion polymerization and embedded into the pNIPAM network. When coordinating the trivalent lanthanide ions to the nanoparticles, a luminescent thermoresponsive polymer with a tunable temperature-responsive range was obtained. The detection principle of this nanothermometer is based on the synergistic effect between the thermoresponsive behavior of pNIPAM and the temperature-dependent emission characteristics of the lanthanide ions. As the temperature increases, the thermoresponsive behavior of pNIPAM causes its volume to shrink, which enhances the red emission of the Eu^3^⁺ complex, while the Tb^3^⁺ complex exhibits different fluorescence changes. By monitoring the ratio of these two emission intensities, the precise quantification of temperature variations can be achieved. The nanoparticle system demonstrates high relative thermal sensitivity, excellent repeatability, and a reversible response to temperature changes. These features remain stable in aqueous solutions, making this ratiometric nanothermometer highly suitable for applications requiring precise temperature measurements in biological systems.

## 4. Conclusions

In this review, we have highlighted the unique characteristics and applications of ratiometric lanthanide coordination polymers (Ln-CPs) in sensing and detection (Table 1). These materials combine the exceptional optical properties of lanthanide ions, such as narrow-band emission, long fluorescence lifetimes, and excellent photobleaching resistance, with the structural flexibility and tunability of coordination polymers. As a result, they have emerged as powerful tools in a wide range of fields, including biological and chemical sensing, environmental monitoring, and medical diagnostics. The ratiometric fluorescence mechanism of Ln-CPs offers a self-calibrating and highly reliable method for precise detection. By utilizing two distinct emission bands, one as a reference signal and the other as a sensing signal, Ln-CPs can accurately measure target analytes, even in complex environments. The relative intensity ratio between these two emission bands changes in response to the concentration of the target analyte, allowing for real-time monitoring with minimal interference from environmental factors such as temperature, light intensity, and background noise. The versatility of Ln-CPs is further enhanced by their ability to incorporate a second luminescent center, such as organic molecules or quantum dots. This adds complexity and sensitivity to the system, enabling the design of multi-channel sensors capable of detecting a wide range of targets, from small ions to complex biomolecules.

Despite significant progress in the development of ratiometric Ln-CP sensors, several challenges remain. These include the limited range of detectable target analytes, and the need for further improvements in the uniformity and spectral performance of the materials. Future research could focus on the following areas:

(1) Further exploration of the coordination and self-assembly patterns of lanthanide metal ions with different ligands to optimize the preparation process and obtain ratiometric Ln-CPs with a controllable and uniform morphology.

(2) The use of long-lifetime reference luminescent molecules, such as phosphorescent species. An important advantage of lanthanide luminescent materials is their constant fluorescence lifetime, which can be utilized in a time-gated mode to avoid background interference. However, current reference luminescent molecules generally have short fluorescence lifetimes, limiting the full potential of lanthanide-based materials. Future studies could explore the use of long-lifetime luminescent species, thereby enabling the construction of dual-phosphorescent ratiometric detection systems.

(3) Current reference luminescent species typically emit at shorter wavelengths. Future studies could explore systems with longer wavelengths or even near-infrared emissions to avoid background interference from complex samples.

(4) The consideration of up-conversion or two-photon luminescent materials, which would allow the use of near-infrared excitation light, benefiting from its strong penetration ability.

(5) The further exploration of the luminescence mechanisms of Ln-CPs, utilizing the energy transfer processes between the components that make up the Ln-CP, to develop a wider variety of ratiometric luminescent detection systems.

With continued advancements in material design, synthesis techniques, and a deeper understanding of the underlying mechanisms, ratiometric Ln-CPs are expected to play a critical role in a broader range of applications, providing innovative solutions in biosensing, environmental monitoring, and beyond.

## Figures and Tables

**Figure 1 molecules-30-00396-f001:**
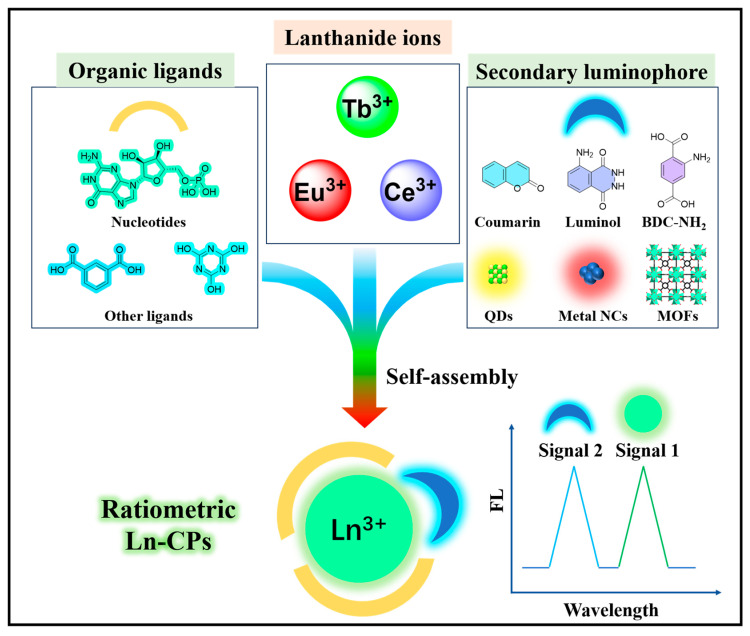
Schematic diagram of the structure of ratiometric Ln-CPs.

**Figure 2 molecules-30-00396-f002:**
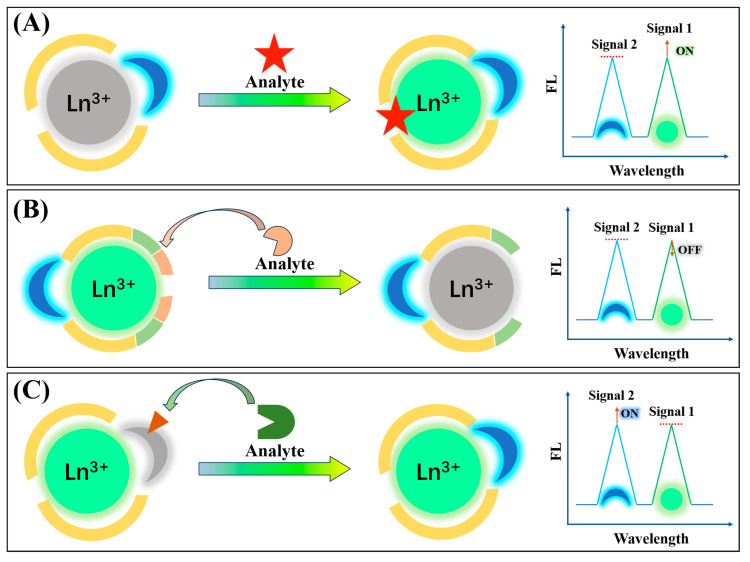
Schematic diagram of the sensing mechanism of ratiometric Ln-CPs sensors. (**A**) Sensitization effect of the target analyte on the lanthanide ion; (**B**) Reaction between the analyte and the ligand or guest molecules in the Ln-CPs; (**C**) Reaction between the analyte and the guest molecule (second luminescent center).

**Figure 3 molecules-30-00396-f003:**
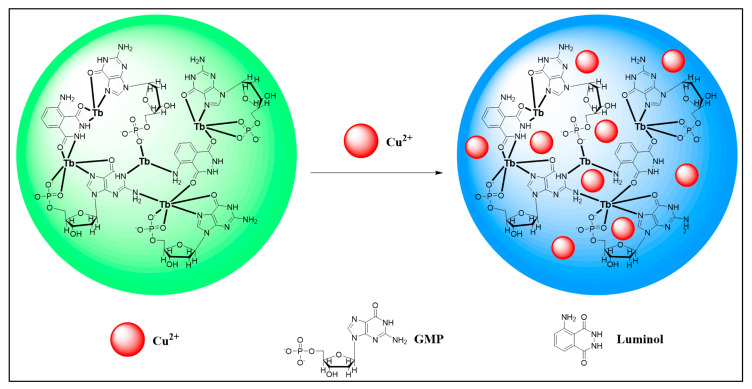
Schematic illustration of the preparation of dual-emission GOx&CDs@AMP/Tb-CPBA coordination polymers and their working principle for ratiometric glucose detection. Reproduced with permission [65]. Copyright 2018 American Chemical Society.

**Figure 4 molecules-30-00396-f004:**
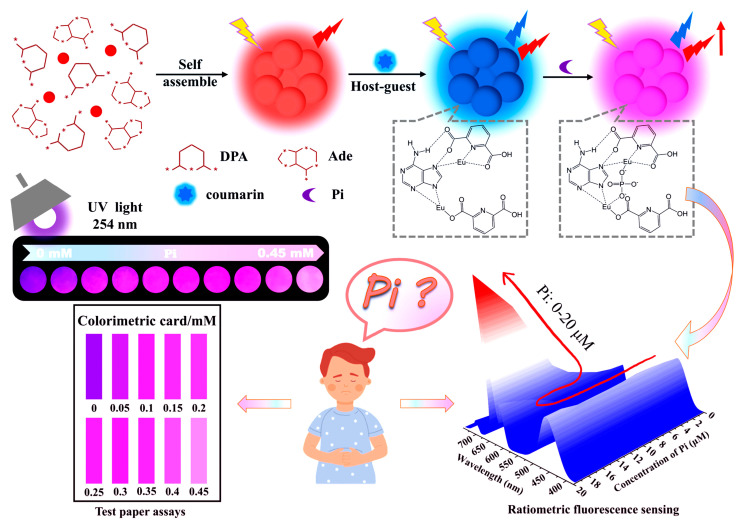
Schematic diagram of the preparation of COU@Eu-ICPs and its mechanism for phosphate ion detection. Reproduced with permission [82]. Copyright 2022 Elsevier.

**Figure 5 molecules-30-00396-f005:**
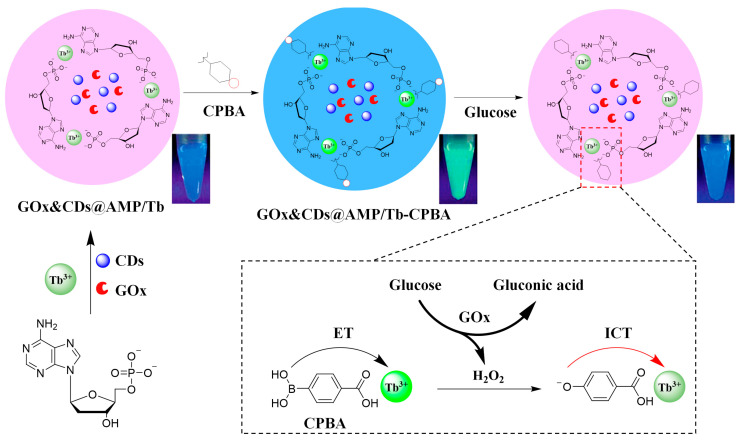
Schematic illustration of the preparation of dual-emission GOx&CDs@AMP/Tb-CPBA coordination polymers and their working principle for ratiometric glucose detection. Reproduced with permission [93]. Copyright 2017 Royal Society of Chemistry.

**Figure 6 molecules-30-00396-f006:**
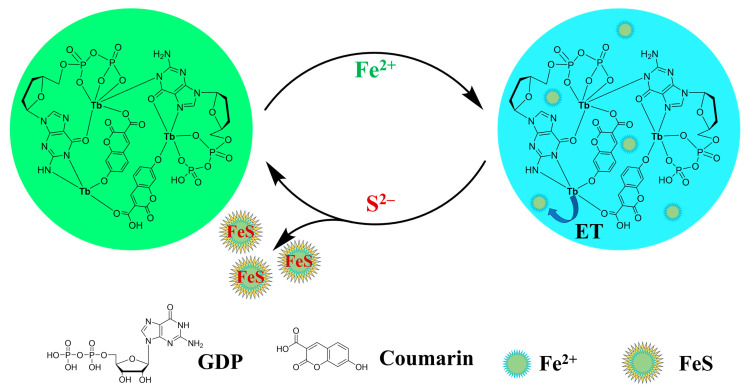
Schematic diagram of the preparation of COU@Eu-ICPs and its mechanism for phosphate ion detection. Reproduced with permission [103]. Copyright 2021 Elsevier.

**Figure 7 molecules-30-00396-f007:**
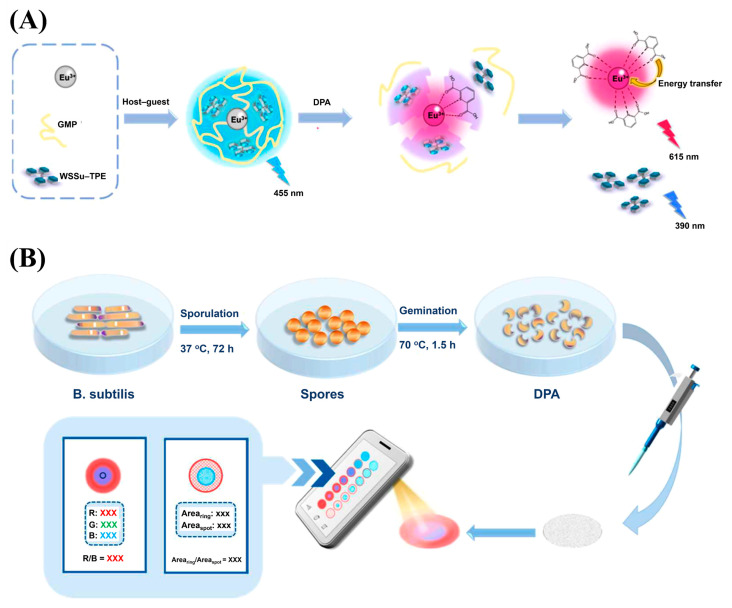
(**A**) Schematic diagram of the preparation of TPE-TS@Eu/GMP ICP nanoparticles and their response mechanism to DPA. (**B**) Schematic diagram of the point-of-use application. Reproduced with permission [121]. Copyright 2020 American Chemical Society.

**Figure 8 molecules-30-00396-f008:**
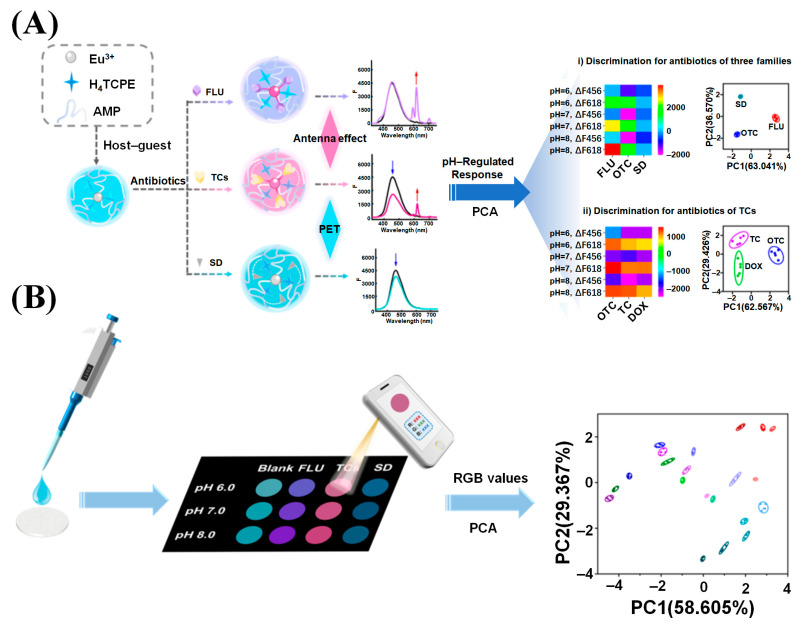
(**A**) Schematic representation of the composition of the H_4_TCPE@Eu/AMP ICP sensor array and its process for antibiotic analysis. (**B**) Schematic illustration of paper-based analysis for on-site applications, assisted by a smartphone. Reproduced with permission [135]. Copyright 2021 American Chemical Society.

**Figure 9 molecules-30-00396-f009:**
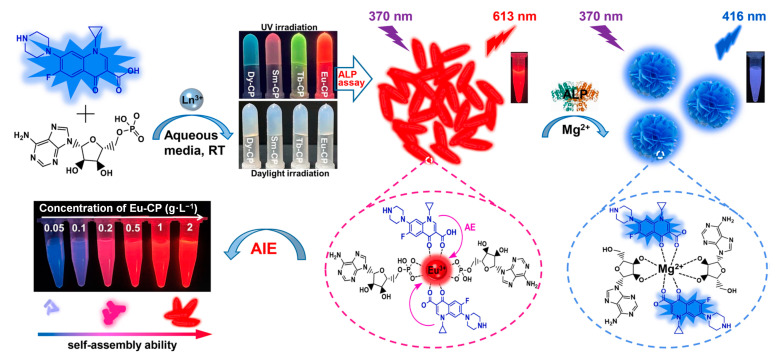
Schematic diagram of the preparation of Ln/AMP/CIP and its mechanism for ALP detection based on the aggregation-enhanced antenna effect. Reproduced with permission [146]. Copyright 2023 Elsevier.

**Figure 10 molecules-30-00396-f010:**
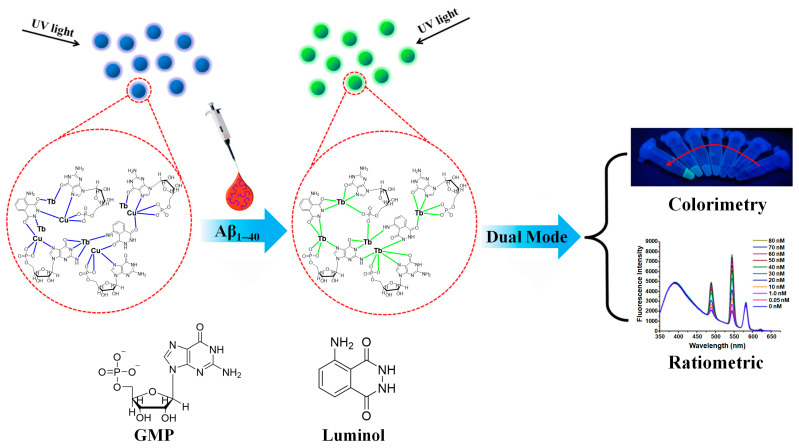
Schematic diagram of the sensing mechanism of luminol-Tb-GMPCu for amyloid β-peptide. Reproduced with permission [158]. Copyright 2022 Elsevier.

**Figure 11 molecules-30-00396-f011:**
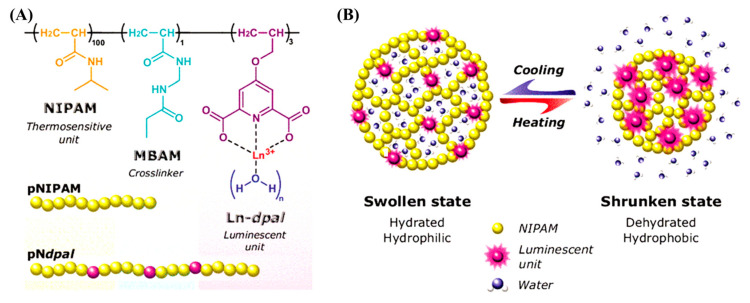
(**A**) Schematic structure of pNIPAM nanoparticles grafted with lanthanide complexes. (**B**) Schematic diagram illustrating the temperature-responsive mechanism of the pNIPAM nanoprobe. Reproduced with permission [170]. Copyright 2020 Royal Society of Chemistry.

**Table 1 molecules-30-00396-t001:** A summary of luminescent lanthanide infinite coordination polymers for ratiometric sensing applications.

Analyte	Ln-CPs Probe	Ratiometric Signal Moieties	Dynamic Range	Detection Limit	Ref.
Fe^2+^	AMP/Tb/Phen/CDs	Tb/CDs	2–500 nM, 0.5–130 μM	2 nM	[61]
Fe^2+^/Fe^3+^	AMP/Tb/Phen/CDs	Tb/CDs	Fe^2+^: 0.05–130 μM Fe^3+^: 0.1–80 μM	Fe^2+^: 23 nM Fe^3+^: 88 nM	[62]
Cu^2+^	GMP/Tb/Luminol	Tb/Luminol	0.01–80 μM	4.2 nM	[65]
Ag^+^	GMP/Tb/Luminol	Tb/Luminol	0.1–6 nM, 6–100 μM	65 nM	[68]
Hg^2+^	GMP/Tb/Luminol	Tb/Luminol	5 nM–130 μM	1.3 nM	[73]
Hg^2+^	Phen/Tb/Luminol	Tb/Luminol	0.1–30 μM	3.6 nM	[75]
Hg^2+^	AMP/Ce/Tb/Coumarin	Tb/Coumarin	0.08–1000 nM	0.03 nM	[72]
Hg^2+^	IPA/Eu/Luminol	Eu/Luminol	0.05–20 μM	13.2 nM	[74]
PO_4_^3−^	AMP/Tb/Lys	Tb/Lys	0.1–6 μM	0.08 μM	[83]
PO_4_^3−^	Ade/Eu/DPA/Coumarin	Eu/Coumarin	0.25–5 μM	51 nM	[82]
PO_4_^3−^	DPA/Eu/Luminol	Eu/Luminol	0.1–25 μM	27 nM	[81]
PO_4_^3−^	BDC/Tb/GMP	Tb/GMP	0.5–100 μM	0.13 μM	[79]
PO_4_^3−^	DPA/Tb/BTB	Tb/BTB	0.1–50 μM	25.8 nM	[80]
^•^OH	DPA/Eu/TA	Eu/TAOH	0.8–200 μM	0.5 μM	[92]
H_2_O_2_	GDP/Cu^2+^/Tb/TA	Tb/TAOH	0.01–300 μM	1.62 nM	[91]
H_2_O_2_	AMP/Tb/3-CPBA	Tb/HBA	0.1–60 μM	0.23 μM	[95]
Glucose	@AMP/Tb/Ce/RhB/GOx	Tb/RhB	0.4–80 μM	74.3 nM	[94]
Glucose	GDP/Tb/BDC/GOx	Tb/BDC	0.005–20 μM and 120–500 μM	3.42 nM	[93]
Glucose	AMP/Tb/CPBA/GOx/CDs	Tb/CDs	0.5–300 μM	51 nM	[92]
H_2_S	GMP/Tb/ZIF-8/CDs/Cu	Tb/CDs	0.5–100 μM	0.15 μM	[102]
H_2_S	GDP/Tb/Coumarin/Fe	Tb/Coumarin	0.1–45 μM	73 nM	[103]
Dopamine	GMP/Tb/Cu^2+^/BDC	Tb/BDC	1–400 μM	0.44 μM	[107]
Histamine	HOF/Eu/FITC	Eu/FITC	3.0–52.5 mg·L^−1^	1.6 mg·L^−1^	[110]
Tryptophan	HOF/Eu/COF	Eu/COF	0.03–1 mM	1.24 μM	[114]
DPA	SiO_2_@HPST/Eu/GMP	Eu/HPST	10–60 μM	24.2 nM	[120]
DPA	GMP/Eu/TPE	Eu/TPE	0–40 μM	27 nM	[121]
DPA	Ligand/Tb/Naphthalene	Tb/Naphthalene	0–7 μM	87 nM	[171]
DPA	SiO_2_@DPA@Tb/GMP/Eu	Tb/Eu	0.05–2 μM	7.3 nM	[119]
DPA	AMP/Tb/Luminol	Tb/Luminol	0.01–10 μM	3.4 nM	[118]
CIP	AMP/Tb/Luminol	Tb/Luminol	5 nM–2.5 μM	2 nM	[125]
CIP	GMP/Tb/BDC	Tb/BDC	0.1–10 μM	23.8 nM	[126]
Tc	GMP/Eu/Luminol	Eu/Luminol	0.01–60 μM	3.4 nM	[133]
Tc	GMP/Eu/MoS_2_	Eu/MoS_2_	12.7 nM–80 μM	3 nM	[132]
Antibiotics	AMP/Eu/H_4_TCPE	Eu/H_4_TCPE	1–30 μM	--	[135]
OTc	AMP/Eu/CDs	Eu/CDs	1–100 μM	0.5 μM	[134]
Pefloxacin	GMP/Tb/BSA-AuNCs	Tb/AuNCs	0.01–60 μM	5.37 nM	[127]
ALP	AMP/Eu/CIP	Eu/CIP	0.1–60 U·L^−1^	0.026 U·L^−1^	[146]
ALP	GMP/Ce/Tb	Ce/Tb	0.2–60 U·L^−1^	0.12 U·L^−1^	[140]
ALP	GMP/Tb/Cu NCs	Tb/Cu NCs	0.002–2 U·mL^−1^	0.002 U·mL^−1^	[142]
ALP	GMP/Eu/ThT/Luminol	ThT/Luminol	0.005–60 U·L^−1^	1.7 mU·L^−1^	[147]
ALP	GMP/Tb/ZIF-8	Tb/ZIF-8	0.25–20 U·L^−1^	0.12 U·L^−1^	[143]
ALP	GMP/Tb/CDs	Tb/CDs	0.5–80 U·L^−1^	0.13 U·L^−1^	[145]
ALP	GMP/Tb@GMP/Eu/DPA	Tb/Eu	0.1–300 U·L^−1^	0.08 U·L^−1^	[141]
ALP	GMP/Tb/Coumarin	Tb/Coumarin	0.025–0.2 U·mL^−1^	0.01 U·mL^−1^	[139]
ALP	AAP/Tb/UiO-66-NH_2_	Tb/UiO-66-NH_2_	0.05–0.6 U·mL^−1^	0.018 U·mL^−1^	[144]
ALP	CIP@SiO_2_/Ce/ATP/Tris	Ce/CIP	0.1–20 U·L^−1^	2.5 mU·L^−1^	[148]
AChE	GMP/Tb/GQD	Tb/GQD	0.1–100 U·L^−1^	0.037 U·L^−1^	[152]
Aβ	GMP/Tb/Luminol/Cu^2+^	Tb/Luminol	0.05–80 nM	20 pM	[158]
Aβ	GMP/Eu/CDs/Cu^2+^	Eu/CDs	0.5–100 nM	0.17 nM	[157]
HIV antigen	ATP/Ce/Fluorescein	Ce/Fluorescein	4–28 pg·mL^−1^	1.1 pg·mL^−1^	[96]
Water content	GMP/Ce/DPA/Tb/DPA	Tb/Ce	0.2–90% in ethanol	0.16%	[162]
Water content	TDA/Eu/Coumarin 343	Eu/Coumarin	10–100% in ethanol	0.2%	[161]
pH	CIP/Eu/Tb	CIP/Eu/Tb	3–12	--	[166]
Temperature	pNIPAM/Eu/Tb	Eu/Tb	36–50 °C	--	[170]

## Data Availability

Not applicable.

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
