# Peer review of "Luminescent Lanthanide Infinite Coordination Polymers for Ratiometric Sensing Applications"

_molecules, 2025, doi:10.3390/molecules30020396_

Round 1
Reviewer 1 Report
Comments and Suggestions for Authors
This review article provides a comprehensive summary of the application of luminescent lanthanide infinite coordination polymers (Ln-CPs) in ratiometric sensing. It introduces the basic composition, sensing mechanisms, and advantages of Ln-CPs in ratiometric sensing. By utilizing two distinct emission bands, Ln-CPs offer self-calibration, improving sensing accuracy and reducing interference from environmental factors, making them highly suitable for precise detection in complex environments. The paper then discusses the diverse applications of Ln-CPs in ratiometric sensing, including environmental monitoring, medical diagnostics, and food safety. Furthermore, the review outlines future research directions. Overall, this article provides an in-depth analysis of the applications of luminescent lanthanide infinite coordination polymers in ratiometric sensing, offering insights into future research potential and challenges. It is suggested that the paper be accepted for publication after revisions.
1. Figure 2C appears to be incorrect and should be revised.
2. A table summarizing the content would help improve clarity and visualization.
3. Ensure the correct use of abbreviations and citation standards.
4. Some of the figures lack sufficient clarity and should be improved.
Author Response
Comment 1: Figure 2C appears to be incorrect and should be revised.
Response: Thank you very much for your valuable feedback and comments. We have revised Figure 2C by removing the unnecessary pentagon-shaped graphic that appeared in the original image.
Comment 2: A table summarizing the content would help improve clarity and visualization.
Response: We sincerely appreciate the constructive suggestion from the reviewer. In response, we have summarized the main luminescent lanthanide infinite coordination polymers (Ln-CPs) for ratiometric sensing applications and have added a corresponding table to the manuscript for better clarity.
The new table is titled: “Table 1. A Summary of Luminescent Lanthanide Infinite Coordination Polymers for Ratiometric Sensing Applications.”
Comment 3: Ensure the correct use of abbreviations and citation standards.
Response: We have carefully reviewed all abbreviations in the manuscript and have made every effort to ensure correct usage and adherence to proper formatting standards.
Comment 4: Some of the figures lack sufficient clarity and should be improved.
Response: We have carefully reviewed all the figures in the manuscript and have redrawn certain figures, such as Figure 4 and Figure 5, to improve their clarity.
Reviewer 2 Report
Comments and Suggestions for Authors
This paper provides a systematic review of the applications of luminescent lanthanide-based infinite coordination polymers (Ln-CPs) in ratiometric sensing. It discusses the working mechanisms of ratiometric sensors and highlights the potential of Ln-CPs in real-time monitoring of target analytes, including ions, small molecule biomarkers, drug molecules, and macromolecules. The flexibility of Ln-CPs allows for effective integration of secondary luminescent centers, further enhancing their applications in sensing systems. The authors also look ahead to future research directions, proposing the introduction of long-lived reference luminescent molecules, exploration of near-infrared emission systems, and the development of upconversion or two-photon luminescent materials to improve the design and synthesis of Ln-CPs, thereby expanding their application range in biosensing and environmental monitoring. This is an exciting and promising research field that warrants further attention. The paper is recommended for publication after minor revisions.
1. Avoid redundant definitions of abbreviations in the main text. It is recommended that the authors carefully check and handle the first appearances of all abbreviations.
2. Figure 3 lacks clarity. It is suggested that the authors redraw the figure to improve its readability and effectiveness.
3. Some relevant new literature is not cited in the paper. It is recommended that the authors further search for and include these references to avoid missing important studies.
4. The citation format needs to be further standardized. Specifically, reference 85 is missing page numbers.
Author Response
Comment 1: Avoid redundant definitions of abbreviations in the main text. It is recommended that the authors carefully check and handle the first appearances of all abbreviations.
Response: Thank you very much for your valuable feedback and comments. We have carefully reviewed all abbreviations in the manuscript and have ensured that their first appearances are handled correctly and consistently.
Comment 2: Figure 3 lacks clarity. It is suggested that the authors redraw the figure to improve its readability and effectiveness.
Response: We have redrawn Figure 3 to enhance its clarity.
Comment 3: Some relevant new literature is not cited in the paper. It is recommended that the authors further search for and include these references to avoid missing important studies.
Response: In response to the reviewer’s valuable suggestion, we have conducted a thorough search for relevant recent literature and have added important references to the manuscript.
Comment 4: The citation format needs to be further standardized. Specifically, reference 85 is missing page numbers.
Response: Thank you for your valuable feedback. We have carefully reviewed and standardized the citation format throughout the manuscript. Reference 85 is an online article that has not yet been assigned official page numbers. We have added the DOI number for this reference to ensure completeness. “Wu, X.; Ruan, C.; Zhu, X.; Zou, L.; Wang, R.; Li, G. Copper-doped Lanthanide Coordination Polymers as Luminescent Nanoenzyme for Ratiometric Sensing of H2O2 and Glutathione. J. Fluoresc. 2024, DOI: 10.1007/s10895-024-03659-z.”
Reviewer 3 Report
Comments and Suggestions for Authors
The authors have performed a review of the recent publications on the study of the composition of ratiomeric lanthanide-based coordination polymers (Ln-CPs), the investigation of the mechanisms of their sensing, and as well as works reporting on the applications of such ratiometric Ln-CPs. The authors have paid most attention to the disclosing the main findings in sensing different ions, radicals, inorganic and organic molecules of great importance in biochemistry and medicine, but also devoted a short part of the review to the describing the main mechanisms of the sensing. The conclusions are consistent with the description and discussion in the main text, and provide suggestions for the focus of works on this topic in the near future.
The work is well done, all the references used are appropriate and highlight the progress in this very important field - the sensing, emitting materials for chemical, biomedical, environmental and medical technologies; the graphical material properly illustrates the data discussed in the manuscript. I have read this review with a pleasure.
I believe this review merits the publication in Molecules. Before acceptance I would like to ask the authors to check the numbering of chapters (e.g. line 758: should be 3.3.1, line 815: should be 3.3.2; and line 1198: should be 4) and the references in order to avoid their duplication.
Author Response
Comment 1: I believe this review merits the publication in Molecules. Before acceptance I would like to ask the authors to check the numbering of chapters (e.g. line 758: should be 3.3.1, line 815: should be 3.3.2; and line 1198: should be 4) and the references in order to avoid their duplication.
Response: Thank you very much for your positive feedback and for reviewing our manuscript. We have carefully checked the numbering of chapters and have corrected the errors as pointed out.